# Dissecting the puzzle of tectonic lid regimes in terrestrial planets

Tianyang Lyu [1], Maxim D. Ballmer [2] ✉, Zhong-Hai Li [3] ✉, Man Hoi Lee [1,4] ✉, Jun Yan[5], Benjun Wu[6] & Guochun Zhao [1,7]

The surface tectonic style of rocky planets controls interior evolution, geologic activity, dynamo action, atmospheric composition, and thus, habitability. In the solar system, Earth is unique in exhibiting plate tectonics, but it also displays a protracted history of diverse tectonic regimes. To understand the dynamics of the coupled plate-mantle system, here we explore 2D hemispheric-scale thermochemical mantle convection models with self-consistent magmatism that reach the statistical steady state. By statistical analysis of model predictions, we quantitatively distinguish between six tectonic regimes, including the mobile, stagnant, sluggish, plutonic-squishy and episodic lids. In addition, we discover the episodic-squishy lid regime, characterized by alternating episodes of plutonic-squishy lid and mobile-lid behavior. By mapping out these regimes over a wide parameter space, we constrain the conditions for potential regime transitions during planetary cooling. Based on geological evidence of Earth's tectonic history, our results point to decreasing effective lithospheric strength during planetary evolution, consistent with previously-proposed physical weakening mechanisms. We also suggest an important role of the episodic-squishy lid for early Earth and present-day Venus. Thus, our study helps to understand the tectonic history of terrestrial planets as they cool over time.

Terrestrial planets in our solar system share comparable bulk compositions and internal structures; however, their surface tectonics exhibit notable distinctions[1]. This diversity not only controls mantle thermal evolution, core cooling and thereby dynamo action[2,3], but also impacts surface geomorphology and atmospheric processes[4]. The style of lithospheric dynamics orchestrates the intricate interplay between interior and surface processes, setting the stage for the long-term evolution and potential habitability of terrestrial planets[5].

In the solar system, the Earth stands out as a unique planet with a mobile-lid regime[6,7], characterized by continuous motion, subduction and deep cycling of coherent lithospheric plates. These dynamic processes promote global-scale mass (e.g., volatiles) and energy fluxes. The specific present-day style of mobile-lid behavior is referred to as "plate tectonics"[8]. In the mobile-lid regime, plate motion is largely driven by the negative buoyancy of the subducting lithospheric plates, referred to as "slab pull". Thereby, the mantle is dominantly stirred by the continuous subduction of its own cold thermal boundary layer: the (oceanic) lithosphere.

In contrast, Mars and many other planetary objects, such as Vesta, currently exhibit stagnant-lid behavior[9,10], distinguished by a single

¹NWU-HKU Joint Center of Earth and Planetary Sciences, Department of Earth and Planetary Sciences, The University of Hong Kong, Hong Kong, China. ²Department of Earth Sciences, University College London, London, UK. ³State Key Laboratory of Earth System Numerical Modeling and Application, College of Earth and Planetary Sciences, University of Chinese Academy of Sciences, Beijing, China. ⁴Department of Physics, The University of Hong Kong, Hong Kong, China. ⁵Department of Earth Sciences, Freie Universität Berlin, Berlin, Germany. ⁶School of Earth Sciences and Engineering, Nanjing University, Nanjing, China. ⁷State Key Laboratory of Continental Dynamics, Department of Geology, Northwest University, Xi'an, China. ✉e-mail: m.ballmer@ucl.ac.uk; li.zhonghai@ucas.ac.cn; mhlee@hku.hk

thick and inflexible lithospheric plate without notable surface mobility and tectonics. The episodic-lid regime[9,10], defined by intermittent episodes of sudden mobilization of an otherwise stable stagnant lid, represents a transitional regime that alternates between stagnant and mobile lid episodes. The sluggish-lid regime[1,11,12] has been suggested for early Mars[6,13]. In this regime, typical plate velocities are notably smaller than average velocities in the convecting mantle, and lithospheric deformation is mostly driven by convective traction (basal drag), rather than subduction (slab pull). As the lithosphere is mostly passively dragged into the mantle, convergent zones are not localized at narrow plate boundaries.

In hot planets, such as the early Earth, magmatism becomes an essential factor for interior-exterior heat exchange, thus impacting overall planetary dynamics and tectonics. For example, in the heat pipe mode[14–16], massive extrusive volcanism is an efficient heat transport mechanism. Rapid cooling of dominantly extrusive volcanism results in the formation of a thick and cold crust and lithosphere, similar to the stagnant-lid regime[17,18]. However, the geologic record on Earth indicates that plutonism dominates over extrusive magmatism (i.e., volcanism), independent of the tectonic setting (continents, mid-ocean ridges, hotspots)[19,20]. For realistic ratios of intrusion to extrusion (4:1 to 9:1)[19,21], another mode of planetary tectonics emerges: the plutonic-squishy lid regime[22]. This regime is characterized by thermal weakening of the mid-lithosphere due to magmatic intrusions. Such local weakening promotes tectonic activity such as rifting, delamination and incipient subduction, resulting in regional and intermittent plate-like mobility. However, sustained plate motions and a global network of plate boundaries are precluded by frequent slab breakoffs at convergent zones[23], or even dripping and (sub-)lithospheric detachment[22], in a hot, low-viscosity asthenosphere.

Numerical simulations have significantly contributed to our understanding of the dynamics associated with these different tectonic lid styles. However, a critical gap remains concerning the conditions that lead to these regimes and their transitions to each other, e.g., during planetary thermal evolution. This challenge arises from the stochasticity of the complex plate-mantle system[24]. Previous studies have shown that small changes in initial conditions can lead to different tectonic styles, even with the same boundary conditions and physical properties[24,25]. This leads to a hysteresis of the plate-mantle system, where otherwise identical parameters can result in non-unique tectonic states. In addition, internal and basal heating of the mantle changes over planetary evolution, making it challenging to predict transitions between different tectonic regimes.

To overcome this limitation, we here explore statistical steady-state models of mantle dynamics, melting and lithospheric tectonics. In the statistical steady-state, time-averaged model properties do not undergo any systematic variations (e.g., systematic planetary cooling to adjust to the initial condition). This approach allows us to distinguish tectonic regimes quantitatively (using a detailed statistical method), and robustly map out the parameter ranges over which they occur. In this effort, we also discover a tectonic regime, the "episodic-squishy lid". Most importantly, we constrain the relationship of all tectonic regimes relative to each other, with implications for the tectonic history of rocky planets.

## Results and discussion
### Tectonic regimes
In this study, we explore 2D hemispheric-scale thermochemical mantle-convection models with mantle melting and crustal differentiation using the code StagYY[4]. All model cases were run for 10 Gyrs to reach the statistical steady-state. We systematically explore the parameter space by varying core-mantle boundary (CMB) temperature, internal heating rate, upper-mantle activation energy, and effective yield stress. These thermal conditions and rheological parameters are chosen because they vary over geologic time or directly influence the dynamics of the plate-

mantle system, respectively. The assumed magmatic intrusion: extrusion ratio is 9:1. For additional detailed information on the simulation methods and parameter variations in this study, we refer the reader to the "Methods" and Supplementary Information.

In our models, we find six distinct regimes of surface tectonics, five of which have been previously described: mobile lid, sluggish lid, episodic lid, plutonic-squishy lid, and stagnant lid. In addition, we discover a tectonic regime with alternating episodes of plutonic-squishy lid and mobile-lid behavior, which is here defined as the 'episodic-squishy lid'. Representative cases for each of these tectonic regimes are shown in Fig. 1.

We categorize the tectonic regime of each of our cases using quantitative model predictions. By analyzing key parameters such as plateness, surface mobility, root-mean-square surface velocity, average mantle temperature, and lithosphere thickness (Fig. 2), we can effectively discriminate between regimes. Plateness quantifies the localization of surface deformation, with a value of 1.0 indicating a perfectly rigid plate with deformation occurring exclusively along narrow plate boundaries, and a value of 0.0 indicating distributed deformation as in isoviscous convection[4,22]. Surface mobility is defined as the ratio of root-mean-square surface velocity to root-mean-square velocity of the convecting mantle, reflecting a normalized measure for plate speeds[4,22].

The mobile lid regime (Fig. 2a) is characterized by consistently high mobility and plateness, and high surface velocities. In this regime, the lithosphere undergoes continuous yielding at convergent zones, sustaining efficient plate subduction and a persistent state of lid fragmentation. Accordingly, plateness values typically range between 0.8 and 1. Similarly, surface mobility consistently exceeds 1, indicating significant surface displacement, largely driven by slab pull. In turn, surface velocity fluctuates between a few cm/yr to several tens of cm/yr. The continuous recycling of cold tectonic plates contributes to low mantle temperatures (2000-2050 K). Lithospheric thickness typically varies between 20 and 200 km due to the cycle of plate aging and thickening away from mid-ocean ridges.

The stagnant lid (Fig. 2b) represents a regime characterized by minimal lithospheric displacement and deformation, with mobility and surface velocity close to zero. This regime implies a non-fragmented lid structure (i.e., single-plate planet) with very low platenesses (<0.2). The stagnant lid exhibits a significantly thicker lithosphere than in the mobile-lid regime, exceeding 200 km. Due to inefficient planetary cooling, mantle temperatures equilibrate at ~2700 K.

In the episodic-lid regime, the plate-mantle system alternates between episodes of mobile-lid and stagnant-lid behavior[24]. In the case in Fig. 2c, the duration of mobile-lid episodes is relatively short, lasting no longer than 0.3 Gyr, and is associated with very high surface velocities and large-scale lithospheric overturns. The stagnant-lid phases typically last from several hundred Myrs up to a few Gyrs. In some episodic-lid cases, the stagnant episodes are shorter than the mobile episodes (see Fig. S1).

The sluggish lid regime (Fig. 2d) is characterized by platenesses close to 1, indicating localized surface deformation, but low surface mobility and velocity. In this regime, the lithosphere is mostly passively dragged into the mantle by convective traction (surface mobility <1) rather than actively subducting under its own weight as in the mobile-lid regime[1,12]. This results in broad convergent zones with wide double-sided subduction zones that are accommodated by V-shaped conjugate plate-scale yielding, while divergent zones (ridges) remain narrow[13] (see Fig. S9). Double-sided subduction is not representative of present-day Earth dynamics. It may be promoted by our relatively simple rheological model and boundary conditions[26]. Slow plate motions yield thick (~170 km) and rigid lithospheres, and mantle temperatures of ~2300 K. During the later stages of the sluggish-lid case presented in Figs. 1–2 (7.4 Gyr onwards), plume-lithosphere interaction results in notable lithospheric thinning. New ridges (and thus young and thin plates) are formed where plumes interact with the

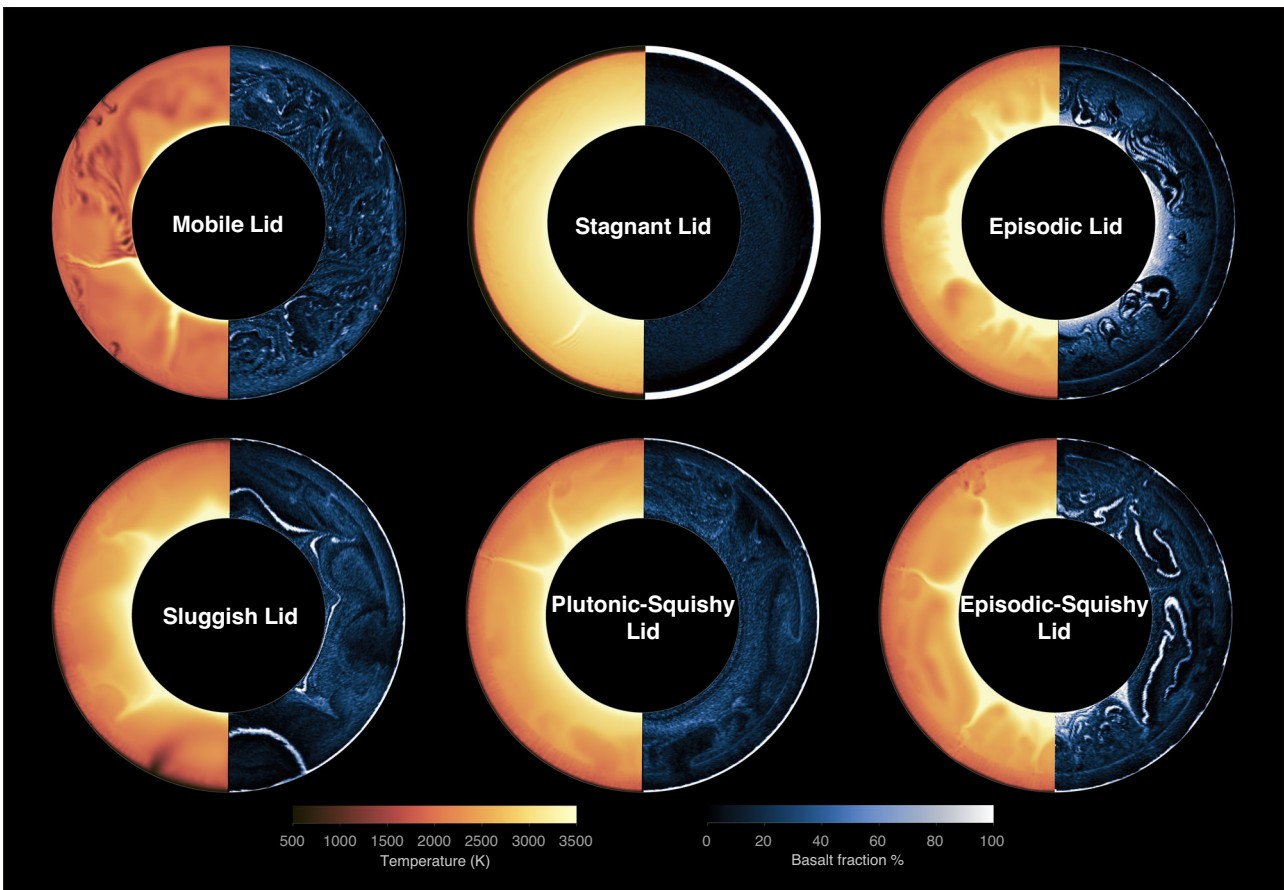

**Fig. 1 | Representative snapshots of the six tectonic regimes.** Snapshots of temperature and basalt fraction for one characteristic case per regime (Mobile Lid: reference1, Stagnant Lid: Rh200cc300, Episodic Lid: Ea250cc80, Sluggish Lid: Rh50cc240, Plutonic-Squishy Lid: Rh75cc300, and Episodic-Squishy Lid: reference5; see Supplementary Data 1).

lithosphere, but the style and number of downwellings do not change. The planet remains in the sluggish-lid regime, and the number of ridges and plates will eventually decrease again due to ridge merging.

The plutonic-squishy lid regime (Fig. 2e) is characterized by transient behavior of tectonic activity. Extended periods with low plateness and very low surface mobility, resembling a stagnant lid, alternate with squishy-lid periods with high plateness (~1) and intermediate mobility (up to ~0.5). During the quasi-stagnant episodes, the lithosphere remains mostly stable, gradually thickening over time, but remaining significantly thinner than in the true stagnant-lid regime. During the squishy-lid episodes, regional deformation occurs due to localized weakening associated with intrusive magmatism[22]. In contrast to the mobile-lid regime, this tectonic activity is accommodated by lithospheric dripping (or slab breakoff during incipient subduction) instead of continuous subduction[22,27]. Accordingly, mantle temperatures are significantly higher than in the mobile-lid regime (but significantly lower than in the stagnant lid).

Finally, the episodic-squishy lid regime (Fig. 2f) alternates between episodes of mobile-lid and plutonic-squishy lid behavior. The typical duration of the mobile-lid stages does not exceed 0.5 Gyr in this episodic-squishy regime. During these stages, lithospheric yielding occurs regionally, accompanied by subduction or delamination, resulting in high surface velocities. In addition to these mobile episodes, the episodic-squishy lid exhibits quasi-stagnant and squishy episodes, which also occur in the plutonic-squishy lid regime (Fig. 3). In the squishy episodes, weak boundaries develop between tectonic blocks due to magmatic intrusions, triggering lithospheric dripping. The resulting return flow promotes extended localized mantle melting

and magmatic weakening (Fig. 3a), eventually leading to subduction initiation and mobile-lid behavior (Fig. 3b). In turn, mantle cooling due to ongoing subduction (Fig. 2f) eventually (i.e., typically after ~0.5 Gyr) leads to a shift into a "quasi"-stagnant lid with a relatively thin lithosphere and therefore finite mobility (Fig. 3c, e). The subsequent rise in mantle temperature eventually promotes melting and intrusive weakening again, either leading to another squishy episode and starting the cycle anew (Fig. 3e, f) or directly switching back to a mobile episode (Fig. 3c, d).

In summary, mobile, stagnant and sluggish-lid regimes exhibit consistent lithospheric behavior (i.e., consistent mobilities and platenesses over time; Fig. 2), whereas episodic, plutonic-squishy and episodic-squishy regimes display highly variable behavior (Fig. 2). In terms of plateness-mobility over time (see scatter plots in Fig. 4), the former three regimes display unimodal distributions, and the latter three regimes display bimodal or trimodal distributions. The unimodal distributions of these former three regimes can be used to define the typical plateness-mobility ranges for these three "base" (mobile, stagnant, sluggish) regimes (Table 1). Fig. S3 shows that these unimodal distributions are robust for all cases in these three base regimes.

Multi-modal distributions for the other three regimes highlight their episodic nature. For example, episodic-lid cases alternate between plateness-mobility values typical for the mobile and stagnant lid. Plateness-mobility values of plutonic-squishy cases overlap with those of the sluggish-base regime during squishy episodes, and with those of the stagnant-base regime during quasi-stagnant episodes. Plateness-mobility values of the episodic-squishy cases overlap with all three base regimes.

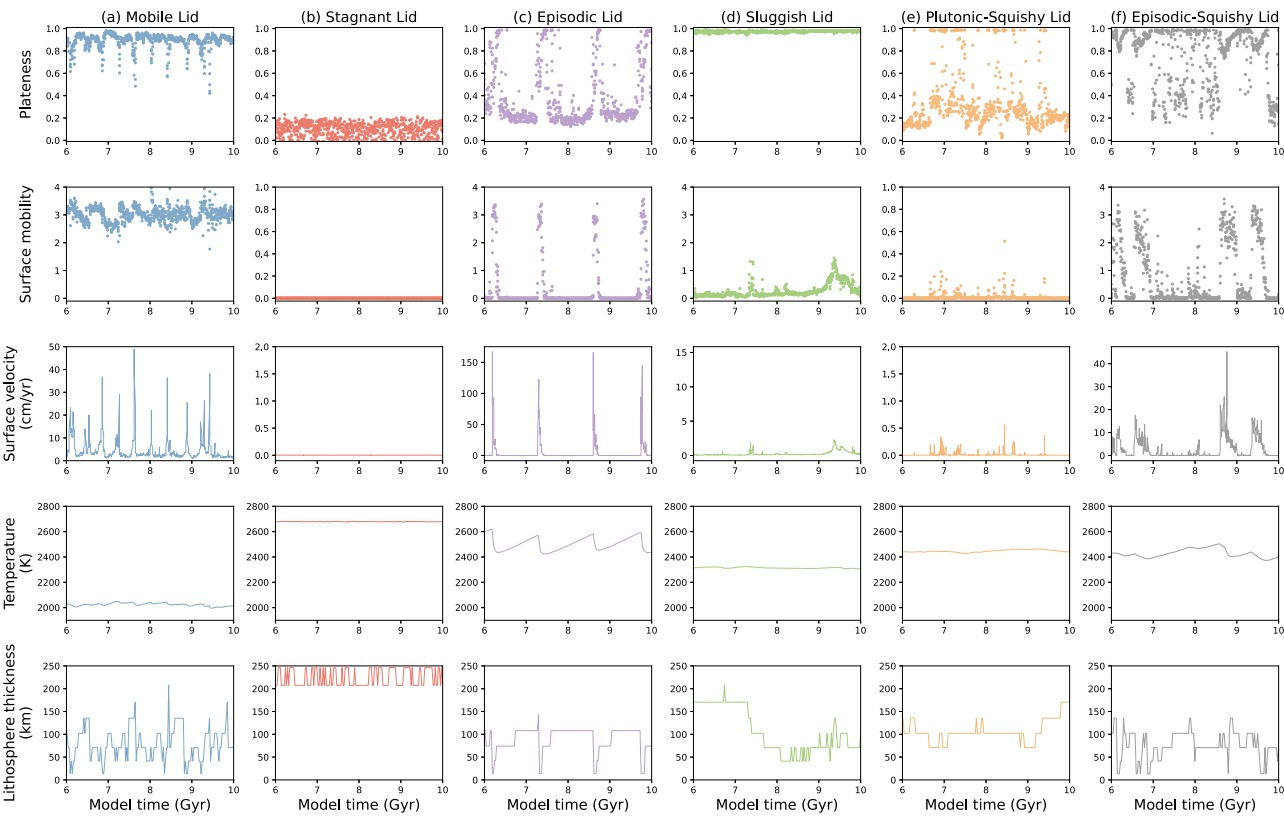

**Fig. 2 | Time-dependent evolution of key parameters in the six tectonic regimes.** Plateness, mobility, root-mean-square surface velocity, average mantle temperature, and lithospheric thickness over model time in the statistical steady state (the last 4 Gyr of a much longer model evolution) for the same six cases as in

Fig. 1: column **a** mobile lid, **b** stagnant lid, **c** episodic lid, **d** sluggish lid, **e** plutonic-squishy lid, and **f** episodic-squishy lid. Note that the vertical axes of mobility and surface velocity are rescaled among cases for visibility.

To quantitatively determine the governing tectonic regime in each of our cases, we quantify the fraction of model time spent in each "base" regime. For example, significant model time is spent in the sluggish/squishy and (near-)stagnant base regimes for the plutonic-squishy lid; most time is spent in the mobile base regime for the mobile lid; etc. (Table 2). This classification can robustly distinguish between all regimes except between the episodic and episodic-squishy lid. Episodic-lid covers all three base regimes during their evolution (Fig. 4c and S3c), because they assume plateness-mobility values typical for sluggish/squishy episodes just after overturns (Fig. 4) due to localized remnant surface deformation (low $M$, high $P$). The episodic lid with planetary-scale overturns can be readily distinguished from squishy episodes in the episodic-squishy regime with regional lithospheric dripping by visual analysis. Such a classification scheme based on model time spent during the three identified base regimes (Table 2) should be widely applicable in global-scale geodynamic modeling studies. We stress, however, that the specific threshold values between base regimes, particularly in terms of mobility (Table 1), are not universally applicable. For example, modifying the mantle viscosity profile will change the ratio between surface and mantle velocities, which defines mobility. While the overall trends and approximate numbers in Table 1 are robust[6,22,28], careful visual analysis of model predictions is required to distinguish between base regimes in any subsequent study. Once base regimes are robustly distinguished, Table 2 can provide a general classification of all six tectonic regimes identified here.

**Parameter study**

We conducted a series of numerical simulations to investigate the effects of mantle thermal conditions and surface yield stress (i.e., parameterized as a cohesion coefficient; see Methods). To study

different planetary thermal conditions, the internal heating rate Rh and CMB temperature $T_{CM}$ are varied in a coupled fashion, varying from hot planets (with Rh = $2.0 \times 10^{-11}$ W/kg, $T_{CMB}$ = 4250 K) to cool planets (i.e., Rh = $5.0 \times 10^{-12}$ W/kg, $T_{CMB}$ = 3750 K). These cases are a proxy for the changing conditions during planetary cooling over time. The effective strength of the lithosphere is explored by varying the surface yield stress, $\sigma_s$. In a simplified Newtonian description of our viscous rheological treatment, we keep the upper-mantle activation energy fixed at $E_{UM}$ = 200 kJ/mol[29]. For each of these cases, Fig. 5a displays the average and standard deviation of plateness and surface mobility (6–10 Gyr); Fig. 5b quantifies the fraction of model time spent in mobile, sluggish/squishy, and stagnant base regimes (Tables 1 and 2).

For low $\sigma_s \lesssim 90$ MPa, and notably independent of mantle thermal condition, plateness and mobility exhibit high average values, but low standard deviations, typical of a mobile lid. Within this regime, tectonic activity intensifies as $\sigma_s$ diminishes, resulting in more mantle cooling due to an increased flux of cold slabs into the mantle. Nevertheless, enhanced tectonic activity (i.e., smaller and faster plates) also leads to more mantle melting (Fig. S2a), notably at mid-ocean ridges.

For hot planets (i.e., Rh = $2.0 \times 10^{-11}$ W/kg, $T_{CMB}$ = 4250 K), models with intermediate surface yield stresses (130 MPa $\lesssim \sigma_s \lesssim$ 180 MPa) display the characteristic features of episodic-lid tectonics, expressed as an intermediate average mobility with high standard deviation (Fig. 5a). In these cases, stagnant episodes are typically short, which ends by overturns with massive melting (Fig. S2a).

For high surface yield stresses ($\sigma_s \gtrsim 180$ MPa), hot planets exhibit a stagnant lid, as indicated by very low plateness and mobility. In this regime, the mantle heats up significantly due to a thick lithosphere and crust, yet only moderate melting occurs (Fig. S2a).

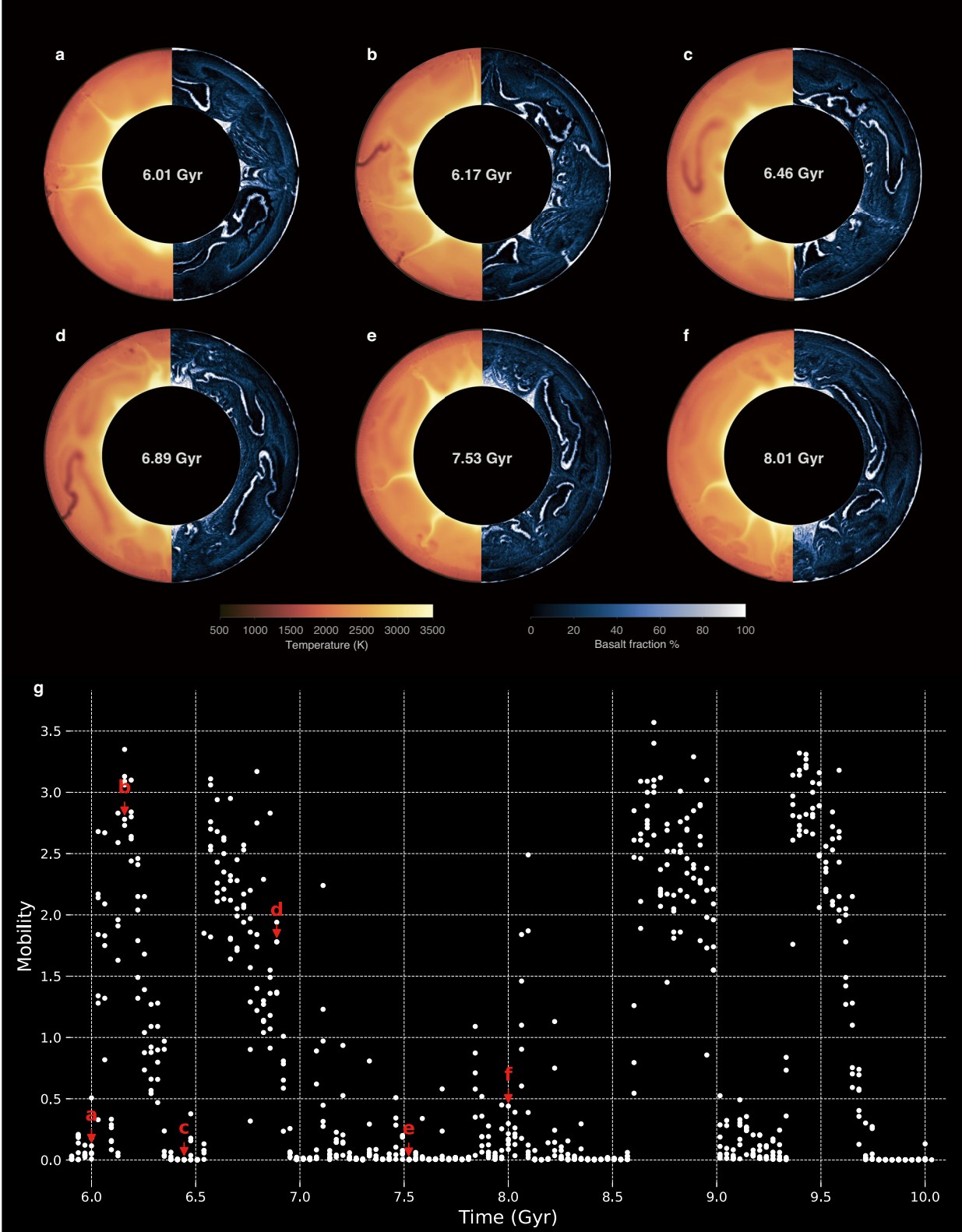

**Fig. 3 | Model evolution and mobility dynamics of the Episodic-Squishy Lid regime. a–f** Snapshots of temperature and basalt fraction for six different snapshots of the Episodic-Squishy Lid model (model ref. 5) as shown in Fig. 2f, where **a** and **f** correspond to plutonic-squishy lid episode, **c** and **e** correspond to stagnant-lid episodes, and **b** and **d** correspond to mobile-lid episodes. **g** Time-dependent mobility of the Reference Model 5 in the statistical steady state, with the six red arrows indicating the points corresponding to the six episodes shown in (**a**–**f**).

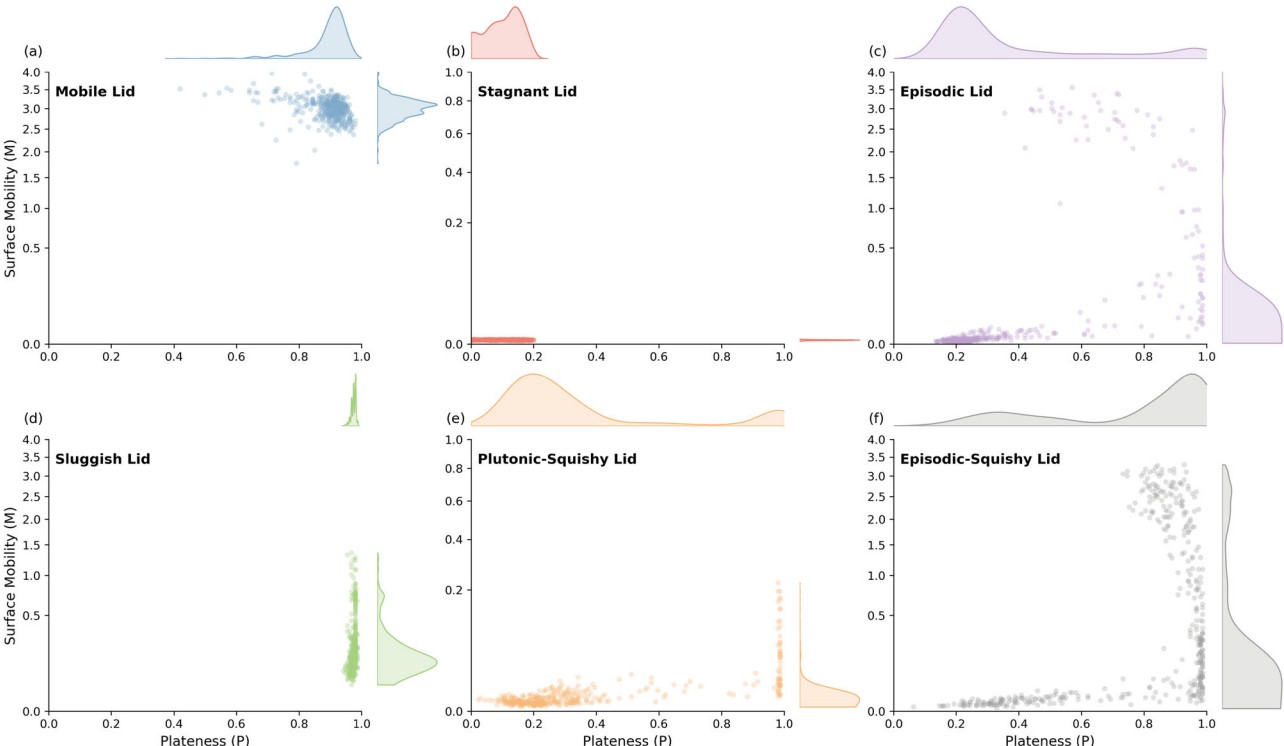

**Fig. 4 | Analyses of plateness and surface mobility in different tectonic regimes.** Scatter plots illustrating the relationship between Plateness (P) and Surface Mobility (M) for six lid modes (as shown in Fig. 2): **a** mobile lid, **b** stagnant lid, **c** episodic lid, **d** sluggish lid, **e** plutonic-squishy lid, and **f** episodic-squishy lid. The data points span the full range of values within 6–10 Gyr time intervals of each case. Kernel density estimates (KDE) for P and M are shown at the top and right of each subplot, respectively, depicting the probability distributions of P and M. The $y$ axis is shown in a square root scale.

**Table 1 | Typical Plateness (P) and Mobility (M) values for each of the three base regimes**

| Base Regime Definition | Plateness (P) | Mobility (M) | Notes |
|---|---|---|---|
| Mobile | P > 0.7 | M > 1.0 | Mobile Lid |
| Sluggish or Squishy | P > 0.7 | M < 1.0 | Sluggish Lid (and squishy episodes of the PSL/ESL regimes) |
| Stagnant (or quasi-Stagnant) | P ≤ 0.7 | M < 1.0 (typically: M << 1.0) | Stagnant Lid (and quasi-stagnant episodes of the PSL/ESL regimes) |

*PSL/ESL* plutonic/episodic squishy lid.

**Table 2 | Classification criteria for each of the six regimes of surface deformation**

| Regime Classification | Model time in mobile base regime | Model time in stagnant base regime | Model time in sluggish/ squishy base regime |
|---|---|---|---|
| Mobile Lid | > 0.6 ** | <0.1 | <0.1 |
| Stagnant Lid | <0.03 | > 0.97 | <0.03 |
| Episodic Lid | > 0.03 | > 0.1 combined * | |
| Sluggish Lid | <0.6 ** | <0.01 | > 0.4 |
| Plutonic-Squishy Lid | <0.03 | » 0.05 | > 0.03 |
| Episodic-Squishy Lid | > 0.03 | > 0.1 | > 0.1 |

Classification criteria are based on the fraction of model time duration within the three base regimes shown in
(*) incipient stagnant phases during the Episodic Lid overlap with the criteria of the sluggish/squishy base regimes in terms of Mobility and Plateness (see main text). (**) In the Sluggish Lid Regime, the average mobility is below 1.0; in the Mobile Lid Regime, it is above 1.0.

For moderate thermal conditions (Rh ≈ $1.0 \times 10^{-11}$ W/kg; $T_{CMB}$ ≈ 4000 K) and $\sigma_s$ ≳ 120 MPa, the variabilities of plateness and mobility are high, while average values steadily decrease with increasing $\sigma_s$. Based on visual inspection and our regime classification (Table 2), these cases shift from the episodic-squishy to the plutonic-squishy lid regime. Mantle temperatures increase as tectonic activity decreases with increasing $\sigma_s$. As mantle melting is promoted by high temperatures as well as tectonic activity, it does not show a clear trend with $\sigma_s$ (Fig. S2a).

Finally, for cool thermal conditions (Rh = $5.0 \times 10^{-12}$ W/kg, $T_{CMB}$ = 3750 K) and $\sigma_s$ ≳ 120 MPa, the average mobility is low, while plateness is consistently high. These characteristics indicate the presence of a sluggish lid, as confirmed by visual analysis. Mantle temperatures are generally low, but systematically increase with increasing $\sigma_s$, attributable to a thicker lithosphere due to reduced surface mobility. For $\sigma_s$ of ~90–150 MPa, a smooth transition between the mobile and sluggish lid occurs (with fluctuations of mobility around 1.0). Figure 5b provides a regime map for this set of models.

Even though the episodic-squishy lid has not been explicitly identified in ref. 22, their Fig. 4 shows high peaks of surface velocities (indicative of the episodic-squishy lid) over a similar parameter range as in this study. The heat-pipe mode does not occur in our models, since we assume a high intrusion:extrusion ratio, but it cannot be excluded, particularly if extrusion is significantly more efficient in the early Earth (or other planets) than the modern Earth.

We also explore the effects of upper-mantle activation energy on tectonic style. In our simplified rheological treatment, the effective upper-mantle activation energy ($E_{UM}$) controls plate-mantle coupling,

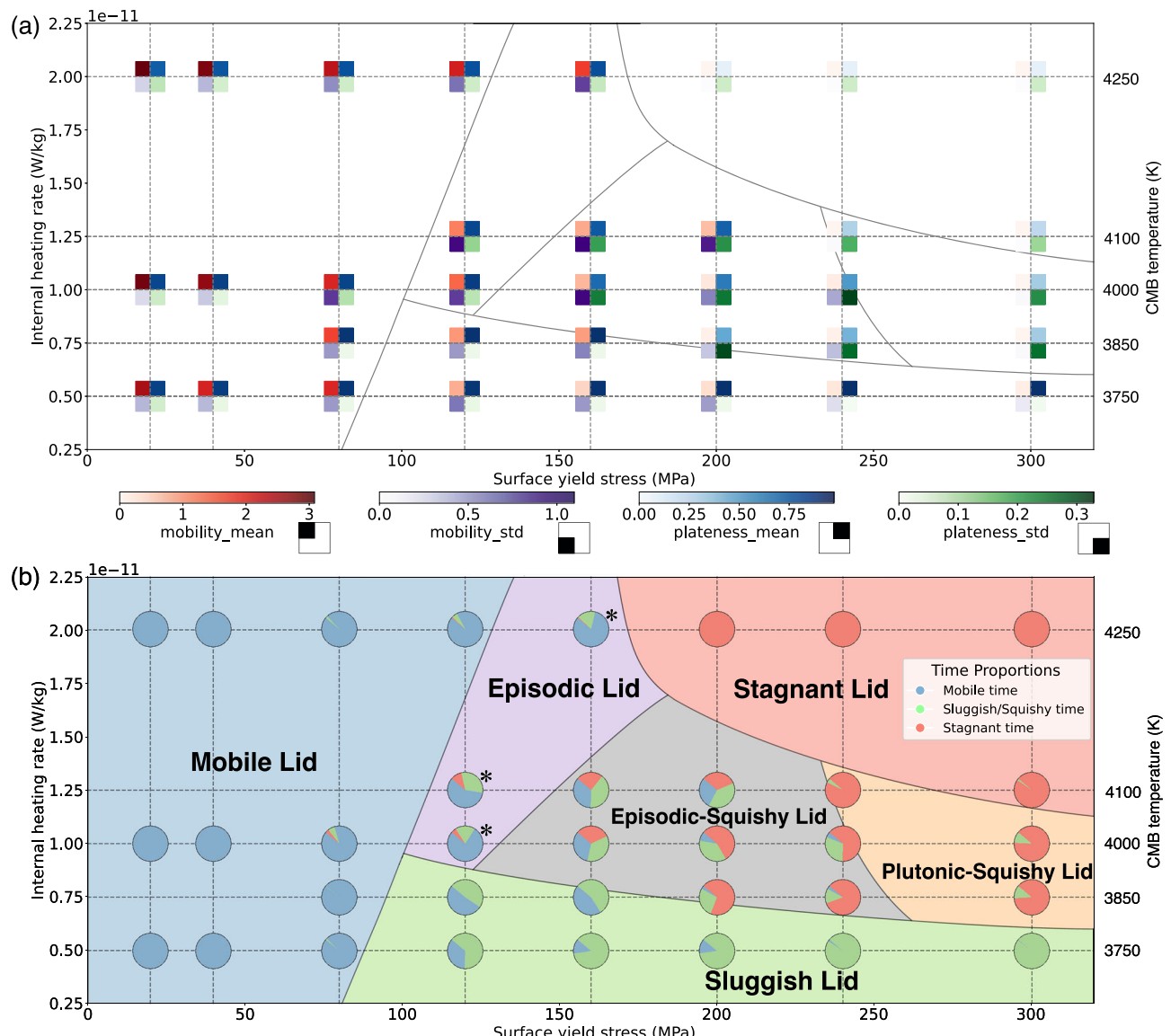

**Fig. 5 | Regime map of tectonic lid dynamics. a** This panel represents the time series for mobility (M) and plateness (P) (as in Fig. 2), summarized as its time-average (or mean) and variability (i.e., standard deviation) for all cases with variable surface yield stress, internal heating rate and core-mantle boundary (CMB) temperature. Note that the CMB temperature (not to scale) is systematically co-varied with the internal heating rate (to scale). **b** Within the same parameter space, this panel shows an overview of regimes (and their boundaries), as inferred from our quantitative criteria (see section Tectonic regimes). These criteria are used to quantify the model time durations of the three base regimes (i.e., mobile, sluggish/squishy, and stagnant) based on M and P (see scatterplot; each pie chart represents the relative contributions of each regime for the corresponding case). (*) incipient stagnant phases during the episodic lid overlap with the criteria of sluggish/squishy base regimes in terms of M and P (see text and Table 2).

thus limiting lithospheric stresses and affecting the tectonic regime. Lower values of $E_{UM}$ can mimic the effects of dislocation creep on global-scale convection, specifically on the viscosity profile of the thermal boundary layers[30]; it can also mimic those of evolving grain-size[31], because grain growth is promoted in hot regions relative to cold regions, trading off with the direct effect of temperature on viscosity (which is captured by $E_{UM}$)[32]. Lower values of $E_{UM}$ enhance plate-mantle coupling, as the viscosity contrast between the lithosphere and asthenosphere becomes smaller, thus promoting mobile-lid behavior (Fig. S1). Lower values of $E_{UM}$ also imply a thinner rheological boundary layer, promoting lithospheric yielding. In turn, high $E_{UM}$ tends to promote a stagnant lid (Fig. S1). The details of our results with variable $E_{UM}$ are reported and explained in the Supplementary Information.

These models with variable $E_{UM}$ also help to understand the behavior of our main model suite in Fig. 5. Stagnant lid cases at high

thermal conditions are promoted by less efficient plate-mantle coupling due to a hot asthenosphere (i.e., analogous to the effects of high $E_{UM}$). With decreasing mantle temperatures (Fig. S2a), plate-mantle coupling increases, promoting the plutonic-squishy lid (with intermittent yielding), and then the sluggish lid (with dominant mantle drag and coherent tectonics) even at the highest yield stresses explored here. In turn, for low yield stresses, mobile-lid behavior (with highly active tectonics) occurs. The transitional episodic and episodic-squishy lid regimes naturally occur in the parameter spaces in between (Fig. 5b).

In addition, lithospheric structure controls the tectonic regime. Large lateral rheological heterogeneity is required to induce localization of deformation and yielding[33]. In our models, localized weakening is predominantly provided by intrusive magmatism. Hot stagnant-lid cases exhibit a thick lithosphere with only moderate magmatism,

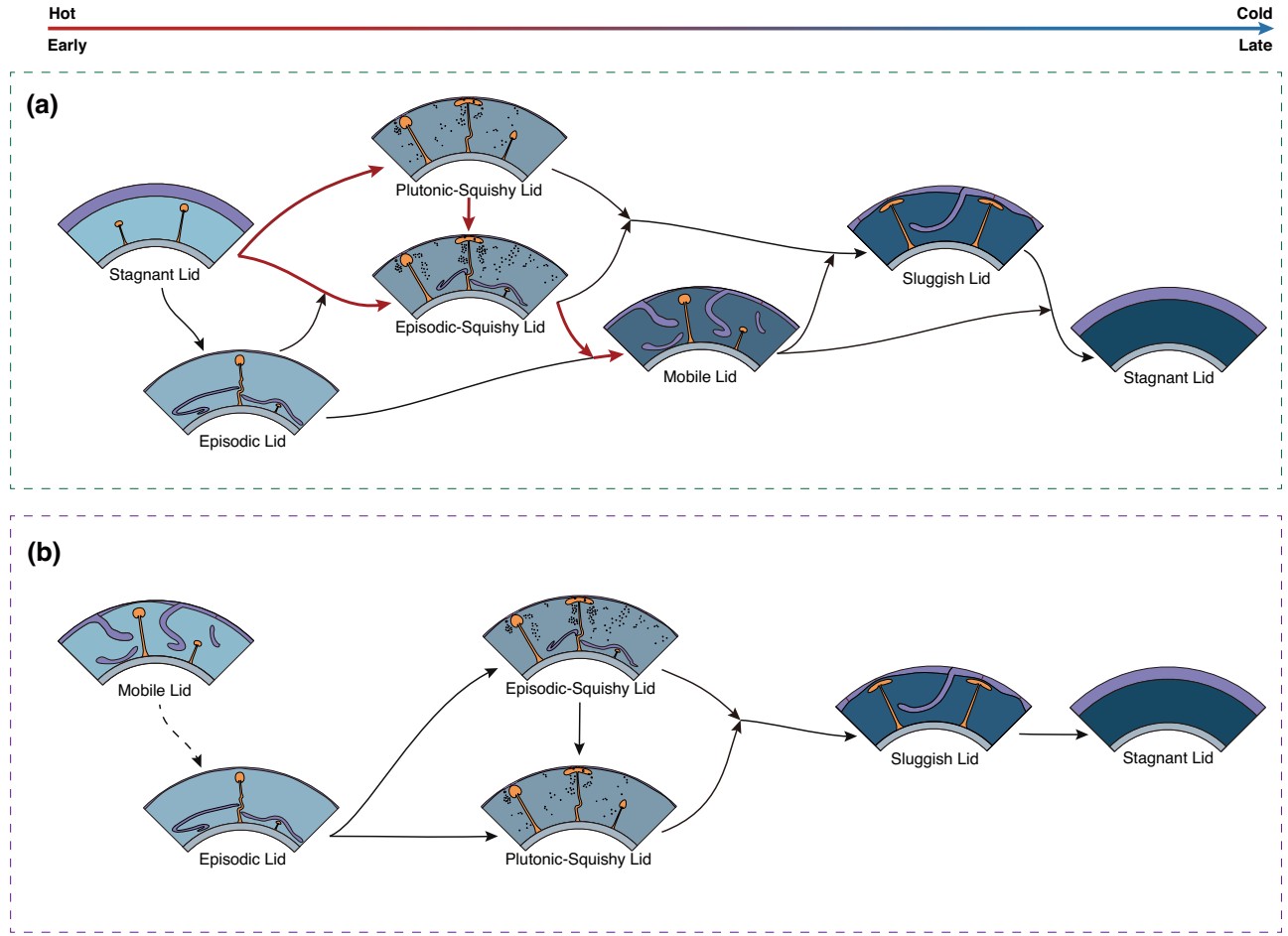

**Fig. 6 | Schematic tectonic evolutionary pathways of terrestrial planets.** Possible tectonic evolutionary pathways of terrestrial planets for **a** decreasing and **b** increasing effective yield strength with planetary cooling. For Earth, decreasing effective lithospheric strength over time can be consistent with a tectonic evolution as highlighted by the red path in (**a**), which is indeed consistent with geologic evidence (see main text). The graphical depiction of layer thickness is not drawn to scale. Moving from left to right, the rocky planets are arranged in order of their transition from a hot (early) state to a cold (later) state.

partly because massive basaltic melt has already been extracted from the mantle early in their evolution. In contrast, warm plutonic-squishy cases display more magmatism (Fig. S2a) and thus a rheologically heterogeneous lithosphere with frequent active (albeit intermittent) yielding events. Cool sluggish-lid cases again exhibit less magmatism, leading to only passive yielding, typically confined to a single subduction zone globally. Even though only minor magmatism occurs in mobile-lid cases, the plate-mantle system with active subduction self-sustains rheological heterogeneity (e.g., thin lithosphere near ridges) and transfers large stresses distally, sufficient to maintain subduction at low yield stresses.

**Planetary tectonic history with evolving lithospheric yield stress**
As in many previous studies[22,34,35], we find that the effective yield stress is a crucial parameter that controls lithospheric strength and its ability to deform under stress (Fig. 5). Our results also highlight the important role of thermal conditions (CMB temperature and internal heating rate). Both these crucial parameters are expected to evolve as the planet cools over time, instead of remaining fixed as in our individual steady-state cases.

Depending on whether the lithosphere effectively weakens or strengthens over time, our models predict diverse trends of tectonic evolution in Fig. 5 (e.g., downward-left or downward-right) as the planet cools over time. If the lithosphere weakens (i.e., $\sigma_s$ effectively decreases) over time, a terrestrial planet may, e.g., evolve from a stagnant lid through an episodic or episodic-squishy to a mobile lid

(Fig. 6a). Conversely, if $\sigma_s$ systematically increases as the planet cools, a planet may evolve from a mobile or episodic lid through an episodic-squishy or plutonic-squishy to a sluggish lid (Fig. 6b).

For Earth, the geologic record points to an evolution as in Fig. 6a., i.e., from the top-right towards the bottom-left of our regime diagram (Fig. 5b). Such an evolution may reconcile the geologic history of Earth, but requires decreasing effective lithospheric yield strengths during planetary cooling. Below, we discuss the geological evidence of Earth's tectonic history and establish potential physical mechanisms that may explain decreasing lithospheric strengths over time.

The dynamics of the present-day Earth are governed by a specific variant of the mobile-lid regime: plate tectonics[8]. However, it is widely accepted that plate tectonics did not operate throughout Earth's history[36], even though the onset time of plate tectonics remains controversial[37]. Geological evidence from Hadean zircons indicates high-temperature melting and limited crustal recycling, suggesting a stagnant-lid regime with inefficient mantle-plate coupling in the early Earth[38,39]. In the Archean, observations such as dome-and-keel structures in cratons (e.g., Pilbara and North China cratons)[40,41] and widespread tonalite-trondhjemite-granodiorite (TTG) rocks[42] point to mostly vertical tectonics with rather inefficient lateral motion[41], and extensive mantle melting[43] that promotes melt intrusion into the lithosphere, widespread delamination as well as crustal dripping[44]. Accordingly, the Archean lithosphere may have been divided into coherent blocks separated by deformable and weak boundaries (i.e., high plateness), but low mobility. This interpretation of the geological

record (e.g., ref. 45) is consistent with the predictions of the plutonic-squishy lid regime (Fig. 2e)[21,22].

Naturally, due to limited geological constraints, we cannot discount the possibility that the early Earth may have alternated between stages with minor mobility and stages of more active plate-tectonic activity, such as in the episodic (-squishy) lid regime[36,39]. If the Earth has evolved through the episodic-squishy lid regime during the Archean and/or early Proterozoic (central path in Fig. 6a), multiple onsets of plate tectonics may be consistent with the diverse interpretation of the geological record[37]. Our episodic-squishy cases demonstrate that switches between planetary tectonic regimes can occur on a timescale of ~0.5 Gyr (Fig. 3).

It is noteworthy that in all these possible scenarios, the geological record points to a trend with increasing mobility over time. Once started, mobile-lid behavior likely evolved from a regime with short-lived subduction and frequent slab breakoff[27,46] to modern-style plate tectonics with sustained subduction zones[8]. This is supported by the transition from Archean vertical tectonics to Proterozoic evidence of horizontal motion, such as ophiolites and blueschists, indicating sustained subduction by ~2 Ga[37]. According to our results (Fig. 5), such increasing tectonic mobility over time, e.g., from a stagnant and/or (episodic/plutonic) squishy lid to a mobile lid, requires that the effective lithospheric yield stress decreases over time.

Various physical mechanisms may account for an effectively decreasing lithospheric strength over time, i.e., with planetary cooling. On the one hand, if the strength of the lithosphere is limited by ductile deformation, the accumulation of damage can control its rheology[47]. As the lithosphere is initially deformed, dynamic recrystallization can lead to localized weakening through grain size reduction (i.e., driven by deformational work), micro-cracks or defects[48,49]. Such localized "damage" builds up over time, effectively decreasing lithospheric strength. Lithospheric heterogeneity—such as due to continental growth[8,50,51] or the formation of large impact basins[52]—may control how damage nucleates and further evolves. Since damage competes with grain growth (or other diffusive "healing" mechanisms), weakening becomes more efficient with (mid-)lithospheric cooling[53,54]. Weakening upon cooling is perhaps counterintuitive, but well explained by the thermal control on the balance between damage and healing[47], as well as by the dominant role of grain size for viscosity[32].

On the other hand, plate deformation may be limited by brittle instead of ductile deformation, at least in the mobile lid. Plate tectonics is critically accommodated by subduction zones, and deformation along the subduction interface is mostly sustained by brittle deformation, as indicated by plate-boundary seismicity reaching >200 km depth[55]. Brittle deformation is limited by fracture propagation, not by static fracture initiation[56]. At temperatures below 600–700 °C[57], fracture propagation is assisted by dynamic frictional instabilities, effectively reducing the frictional strength along the fault[56]. In turn, these instabilities are inefficient at high temperatures, consistent with the distribution of seismicity on Earth, and thus provide another mechanism for effectively decreasing lithospheric strengths during planetary cooling. However, while comparison of our model predictions with the geologic record indeed supports lithospheric weakening over time, it does not provide direct evidence for any of these mechanisms. Alternative mechanisms for progressive weakening involve serpentinization along faults, and lubrication of the subduction interface by hydrous sediments[8,58].

These mechanisms for decreasing lithospheric strength over time indicate a possible evolutionary path of planetary tectonics from the top-right toward the bottom-left in our regime diagram (Fig. 5b). To end up in the mobile-lid regime, such paths tend to transit through the plutonic- and/or episodic squishy lid regime, consistent with geological evidence for the Archean. Accordingly, our model predictions provide further support that these regimes are relevant for the tectonic history of our planet. Very early (Hadean) tectonic regimes

remain poorly constrained geologically, but a "hot" stagnant lid with efficient mantle-plate decoupling is consistent with our results. In the future, the cooling Earth is predicted to evolve into a sluggish lid. With further cooling (i.e., at lower temperatures than shown in Fig. 5b), we expect that terrestrial planets evolve into a "cold" stagnant lid as the ultimate tectonic regime[36], because of a thick lithospheric boundary layer, and thus a strongly reduced mantle Rayleigh number[59] (as is the case for present-day Mars and the Moon).

However, it is not obvious that our steady-state models can be directly applied to the transient evolution of real planets, especially during their early, thermally active stages. To explore the relevance of transient and/or delayed regime transitions, we continued to run some of our steady-state models (i.e., beyond 10 Gyr model time) with an adjusted surface yield stress, $\sigma_s$. While abrupt increases of $\sigma_s$ do not always immediately impact the tectonic style (corresponding regime transitions are delayed by up to ~1.5 Gyr), abrupt decreases of $\sigma_s$ almost instantaneously result in the expected regime transitions (see Text S4 and Fig. S7). Thus, our regime diagram (Fig. 5b) can indeed be applied to Earth's transient evolution, at least for constant or decreasing $\sigma_s$ over time. These results emphasize the robustness of our regime map, and also point to the limited effects of hysteresis in our models, likely due to the regulating effects of mantle melting and crustal emplacement on the lithosphere-mantle system (see Text S4).

Along these lines, our results provide a predictive framework for the evolution of Earth-like planets. While Venus exhibits similarities in size, composition, and internal structure to Earth, its surface temperature is significantly higher. The tectonic regime on present-day Venus remains controversial, and there is no reliable evidence for the prevalent tectonic regime on early Venus due to tectonic and/or volcanic resurfacing ~300 Myrs ago[60,61]. Thus, it remains challenging to evaluate Venus' tectonic evolution (Fig. 6). Lateral motion of regional lithospheric blocks[62], and evidence for plume-lithosphere interaction and/or regional plume-induced subduction, are consistent with the plutonic-squishy lid regime on present-day Venus[63,64]. The alignment between model-predicted structures of plume-lithosphere interaction and the observed morphology of coronae and novae supports this hypothesis[64,65]. On the other hand, Venus' highlands (tesserae) display evidence for large-scale convergent tectonics, possibly related to broad downwelling zones as in the sluggish-lid regime[66]. However, there is no evidence of global-scale ridge systems as characteristic of the sluggish-lid (or, for that matter, the mobile-lid) regime. The results of this study emphasize the similarity between the plutonic-squishy lid regime (or at least of its squishy episodes) and the sluggish-lid regime, as well as their proximity in the regime diagram. Even though our models are not directly applicable to Venus, they point to the possibility of a hybrid (or combined) squishy-sluggish regime.

Another key observation involves the nearly uniform crater distribution and the apparent global surface age of ~300–500 Myr[61]. This observation has been interpreted either in terms of equilibrium or catastrophic resurfacing[67]. In the latter case, the crater distribution would point to at least one (or several) episodic overturn(s) in Venus's history. In light of the evidence discussed above of a weak ("squishy") lithosphere, such episodic overturns may imply an episodic-squishy lid regime on present-day Venus. Indeed, our predictions for the episodic-squishy lid regime may explain brief intervals of enhanced block motion[45,62] with long quiescent periods, offering a simple explanation for both the cratering record[61] and the lack of coherent, global-scale plate boundaries.

The tectonic history of early Venus is largely unconstrained, yet it is crucial for understanding the planet's divergent atmospheric evolution[68,69]. Previous numerical studies[70,71], suggest that early Venus may have been locked in a stagnant lid regime due to its higher surface temperatures, generally consistent with our results. However, it is not obvious that the subsequent evolution of Venus would have been dominated by decreasing effective lithospheric strength over time, as

discussed above for Earth (Fig. 6a). On Venus with its consistently high surface (and hence mid-lithospheric) temperature, weakening from damage accumulation (e.g., via grain-size reduction) would be suppressed by efficient thermal healing (e.g., via grain growth)[47]. Weakening due to dynamic friction instability is also not expected to be efficient[57]. Consequently, Venus's lithosphere may have followed a path of near-constant, or even increasing lithospheric strength (Fig. 6b) over its (early) history[60]. Such paths are consistent with a transition from a stagnant to an episodic-squishy lid over time.

In summary, our study provides a comprehensive classification of tectonic regimes for terrestrial planets based on key model outputs. We identify six distinct tectonic regimes, including a regime termed the episodic-squishy lid, which may be applicable to Archean Earth and present-day Venus. The relevance of the episodic-squishy lid is indicated by its central location (covering a rather large swath of the parameter space) in our regime map, and is also consistent with geological (Earth) and morphological (Venus) evidence. Our results highlight the importance of lithospheric strength, mantle temperature, and plate-mantle coupling in controlling the tectonic behavior of a planet. While future work is needed to explore the parameter space further, our study sheds light on the tectonic evolution of terrestrial planets and can guide future observational studies and modeling efforts, including those of Earth-like exoplanets. As the tectonic evolution of Earth remains poorly constrained, our regime map can help to identify (or rule out) potential historical tectonic transitions.

## Methods

### Mantle convection modeling
Mantle convection is simulated in 2D half-annulus geometry employing the hybrid finite-difference/finite-volume code StagYY[4]. The governing equations for mass, momentum and energy are solved for a compressible fluid, assuming an infinite Prandtl number. The numerical model used here is based on the one described by ref. 35, with the main difference being the parameters used for mantle rheology, internal heating rate, and the thermal condition at the CMB. The values used for the physical parameters in this study are listed in Table S1. The numerical model domain is resolved by 256 (horizontal) × 64 cells (radial), in which ~0.5 million Lagrangian particles (i.e., tracers) are advected. We have also explored some cases with higher resolution and found that our main conclusions remain robust, with detailed analysis provided in the Supplementary Information. To reduce computational costs and enable exploration of a wide parameter space, we have adopted a resolution of 256 × 64 cells in the parameter study below.

### Rheology
We employ a temperature- and pressure-dependent Arrhenius law to simulate planetary tectonics:

$$\eta_m = \eta_0 \Delta\eta \exp\left(\frac{E_a + pV_a}{RT} - \frac{E_a}{RT_0}\right) \quad (1)$$

where $\eta_0$ represents the reference viscosity when reference temperature $T_O = 1600$ K at the lithosphere-asthenosphere boundary; $\Delta\eta$ is the viscosity jump associated with various phase transitions (Olivine–Spinel, Spinel–Bridgmanite, Bridgmanite–Postperovskite)[21]; $E_a$ is the activation energy; $p$ is pressure; $V_a$ is the activation volume; $R$ is the ideal gas constant; $T$ is temperature. Different values for $E_a$ and $V_a$ are employed for the upper mantle, lower mantle, and postperovskite layer[72,73] (Supplementary Data 1 and Table S1).

Deformation experiments of olivine aggregates indicate that $E_a$ is ~360 kJ/mol for diffusion creep[72], but lower values can mimic the effects of dislocation creep in the upper mantle[30]. Therefore, $E_a$ in the upper mantle is varied as a free parameter in this study, particularly as it controls the viscosity profile across the lithosphere, and thereby affects tectonic style.

To simulate plate-like tectonic behavior, plastic yielding is incorporated by applying a simplified pressure-dependent yield criterion[10]. Once an effective yield stress, $\sigma_{eff} = \sigma_s + p\mu$, is reached, the viscosity is artificially reduced to:

$$\eta_Y = \frac{\sigma_s + p\mu}{2\dot{e}} \quad (2)$$

where $\sigma_s$ is the effective surface yield stress (or cohesion coefficient), $\mu$ is a pressure-dependent coefficient, and $\dot{e}$ is the second invariant of the strain rate tensor. At yielding, the effective viscosity at any model grid point is then given by:

$$\eta_{eff} = \min(\eta_m, \eta_Y) \quad (3)$$

Note that the surface effective yield stress, $\sigma_s$, is one of the main model parameters explored in our study. The coefficient $\mu$ is systematically co-varied as $\sigma_s$ is explored (see Supplementary Data 1). This plastic rheology treatment is a simplified parameterization that can simulate effective plate-like behavior on a global-scale[4,22,60]. The effects of more complex lithospheric deformation mechanisms are discussed in the main text.

### Radiogenic heating and boundary conditions
In order to systematically study tectonic regimes without the additional complexity of transient behavior, we aim to explore steady-state results of our numerical simulations of thermochemical evolution spanning 10 Gyrs. To approach the statistical steady state, we focus on the last four billion years of model time (i.e., from 6 to 10 Gyr). Steady-state models offer several advantages compared to models with core cooling. They simplify calculations, provide faster convergence rates, allow more robust analysis of mantle convection characteristics, and facilitate comparison with other similar models. These steady-state models assume a constant internal heating rate and temperature at the CMB, $T_{CMB}$. Each combination of these two model parameters is meant to reflect a different stage of planetary thermal evolution (Supplementary Data 1).

During mantle melting, the distribution of heat-producing elements (HPEs) between the solid mantle and the melt is described by the partitioning coefficient $D_{part}$, defined as:

$$D_{part} = \frac{C_{solid}}{C_{melt}} \quad (4)$$

where $C_{solid}$ and $C_{melt}$ represent the concentrations of HPEs in the solid and melt phases, respectively. HPEs, such as uranium, thorium, and potassium, are strongly incompatible elements, meaning they preferentially partition into the melt during mantle melting. In this study, $D_{part}$ is set to 0.01, reflecting this incompatibility. This low value indicates that most HPEs migrate with the melt to the crust, enhancing radiogenic heating in the crust and depleting the mantle.

As for thermal boundary conditions, the top and bottom boundaries are both set to be isothermal, with a fixed surface temperature of 300 K and $T_{CMB}$, respectively. Free-slip and periodic boundary conditions are applied at the surface/CMB and sidewalls, respectively.

In terms of initial conditions, the initial geotherm is set along the 1900 K adiabat with boundary layers of 30 km thickness at the top and bottom. The initial composition is uniform (pyrolytic). We do not impose any initial mantle or lithospheric velocities. The dynamics and tectonic regime evolve fully self-consistently. The initial temperature and viscosity fields are shown in Fig. S6. We have also investigated some cases with different initial potential temperature values and discovered that they do not significantly influence the results (refer to Supplementary Information for further details).

## Composition and melting-induced crustal production

In our planetary models, the composition of mantle materials is parameterized based on mineral physics data[74,75], dividing minerals into the olivine and pyroxene-garnet systems, which undergo different solid-solid phase transitions. This approach, also used in previous studies[17], allows us to capture the complex mineralogy of the mantle in a simplified parameterization (see Table S2). End-member rock types, basalt and harzburgite, are defined as solid solutions of olivine (ol) and pyroxene-garnet (px-gt) systems (with ol:px-gt of 0.75:0.25 and 0:1, respectively). The composition of a parcel of mantle at any given time ranges between these two end-members and is tracked by the tracers. As an initial condition, we assume a uniform pyrolitic mantle composition corresponding to a mechanical mixture of 20% basalt and 80% harzburgite[76].

Changes in composition are primarily driven by melt-induced differentiation. At each time step, the temperature in each cell is compared to the solidus temperature, derived from experimental data[77,78]. If the temperature exceeds the solidus, the appropriate amount of melt is generated. Above a depth of 300 km[79] any melt formed is instantaneously extracted from the mantle, leaving behind a more depleted residue[80]. The extracted melt is moved vertically, partly to the surface ("eruption"), where it efficiently cools and forms the basaltic crust[81], and partly to the base of the basaltic crust ("intrusion"), where it heats up the lithosphere as it solidifies[22]. In our model, we adopt an intrusion-dominated mode, in which 90% of the melt is intruded (i.e., the eruption efficiency is 10%). Such a high intrusion-to-extrusion ratio is consistent with geological constraints[19,20].

## Data availability

The numerical modeling data generated in this study are publicly available in the Zenodo database at https://doi.org/10.5281/zenodo.14601956, and all other relevant data supporting the findings are included in the main text and Supplementary Information.

## Code availability

The code used in this contribution is the convection code StagYY, which is available for collaborative studies from P.J.T. (paul.tackley@erdw.ethz.ch). Figure 1 is visualized using StagLab (www.fabiocrameri.ch/software).

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

## Acknowledgements

Numerical simulations were performed on the clusters of the National Supercomputer Center in Guangzhou (Tianhe-II) and the GeoModeling Cluster at UCAS. Z.-H.L. was supported by the National Natural Science Foundation of China (42225403). G.Z. was supported by Hong Kong RGC grant (JLFS/P-702/24) and GRF grant (17308023). M.D.B. was supported by UK Research and Innovation (NERC grant NE/X000508/1). B.W. was supported by the National Natural Science Foundation of China (42374112). J.Y. was supported by the European Union (ERC, DIVERSE, 101087755) and the German Research Foundation (No. 1324/6-1, SPP1992). T.L. was supported by the Fundamental Research Funds for the Central Universities and a postgraduate studentship at the University of Hong Kong.

## Author contributions

M.D.B. and Z.-H.L. conceived the study. T.L., Z.-H.L., M.D.B., and M.H.L. designed the set of numerical simulations. T.L. conducted the numerical experiments. T.L., M.D.B., B.W., G.Z., M.H.L., J.Y., and Z.-H.L. analyzed the numerical data and interpreted the results. T.L., B.W., Z.-H.L., and M.H.L. wrote post-processing routines. T.L., M.D.B., and Z.-H.L. produced the figures. All authors contributed to writing the manuscript.

## Competing interests

The authors declare no competing interests.
