## [Transparent Peer Review file · Nature Communications]

Dissecting the puzzle of tectonic lid regimes in terrestrial planets

Corresponding Author: Dr Maxim Ballmer

Version 0:

Reviewer comments:

Reviewer #1

(Remarks to the Author)

The ms. reports results of the comprehensive modeling study of the long-term thermo-mechanical evolution of planet with Earth-like parameters. The study uses a similar modeling approach and continues the previous modeling studies by the ETH team (i.e. Lourenco et al., 2018, 2020). New is that the authors investigate effect of different model parameters than it was done in the previous studies and use “quasi-steady state” approximation. In particular they assume that the CMB temperature and internal heat production is fixed and run model till 10Gy to approach “quasi-steady state”. This allows to attribute different tectonic styles to the domains in two parameter space, where internal heat production/CMB temperature is one parameter and effective lithospheric stress (cohesion) another parameter. In addition, they also investigate effect of changing of activation energy in the upper mantle on tectonic regime. I think that all these models are interesting as some theoretical end members, but I doubt that they can be applied to the evolution of the realistic planets. It is well known that internal heat production in planets strongly change during the evolution due to the decay of radiogenic elements and due to their redistribution from mantle to the crust. CMB temperature also is significantly decreasing. Authors claim (lines 333-335) that their models are applicable to transient planetary evolution with reference to their supplement. However, I did not find convincing evidence for that in their supplement information (see also below).

What is likely robust in the models in the ms. is that at the low cohesion the tectonic regime is always mobile with fast moving surface, at high cohesion the regime is low-mobility with low surface velocities and the intermediate mobility regimes correspond to the intermediate values of cohesion. However, this qualitative result is not new and well described in previous studies (e.g. Lourenco et al., 2018, 2020).

The quantitative results of the models in the ms. are unlikely applicable for the planets in the realistic transient regime. For instance, their results do not match with the modeling results by Lourenco et al. (G3 2020), obtained with the same parameters, but for the transient Earth. Compare Fig.3a from the ms. with Fig.3a from Lourenco et al. (2020). At eruption efficiency of 10% average mobility value above 1 in the transient Earth model by Lourenco et al. (2020) is achieved at cohesion lower than 30 MPa although in the ms. that happened at cohesion lower than 100 MPa. That means that the detailed diagrams of tectonic regimes presented in ms. are unlikely qualitatively applicable to the transient planetary evolution.

In the discussion the authors suggest evolution scenarios for the planets in the cases of lithospheric yield stress increasing or decreasing with time. However, these general scenarios are obvious and could be suggested based on previous studies.

Reviewer #2

(Remarks to the Author)

This manuscript presents a large set of numerical convection models, mapping out regimes of global tectonic behavior as a function of yield stress and internal heat production plus core temperature, as well as viscosity activation energy shown in the supplement. The main purpose is to develop quantitative criteria for classifying models into these regimes and show how the regimes are affected by these key model parameters. The paper identifies a new regime, the episodic squishy lid regime, and discusses how planets might evolve over time through the space of different tectonic regimes mapped out. A critical uncertainty is how the yield stress itself might change through a planet's evolution, dictating the path the planet then takes as

it cools.

The paper is well written, provides new insights on mantle convection regimes, and the conclusions are for the most part supported by models presented. There are a few important points that need to be further developed or considered better, but these can be handled with a revision. I think the paper will be suitable for publication after revision.

Main Comments:

1. The paper states on lines 105-107 that issues related to hysteresis and initial conditions are resolved by running the models for ~10 Gyrs to reach quasi-steady-state. While I think the approach of analyzing models in quasi-steady-state is a good one, this does not eliminate issues with hysteresis and initial conditions. In Weller & Lenardic's work, they ran models starting from different initial regimes (stagnant lid versus mobile lid) to quasi-steady-state and still found that in some cases the initial condition dictated the final state. Models with the exact same parameters could end up in different regimes, even when quasi-steady-state is reached, based on starting in different initial convection regimes. So just running models for a long time does not resolve the issues with hysteresis.

I don't think this paper necessarily needs to fully explore hysteresis the way Weller & Lenardic did. But it should at least clearly report the initial conditions for context here. Are the models initially started from a mobile lid or stagnant lid state? What do the initial temperature, viscosity, and velocity fields look like? These are important for then interpreting the quasi-steady-state regimes that develop.

2. The paper doesn't give much physical insight into why the regimes develop in the parameter space shown. I think more should be done here, as the regime diagrams are the key result of the paper. While previous papers have developed scaling relationships for, e.g., convective stress and applied these to explain boundaries between mobile and stagnant-lid regimes, this manuscript has a more comprehensive set of regimes mapped out, and therefore providing physical insight into why the regimes lay where they do would make for a more significant contribution.

For example, in Figure 3b, it is not clear to me why stagnant-lid behavior is only seen at high internal heat rates, rather than being a regime that can always be reached at high yield stresses. Why do low internal heating rates result in sluggish lid behavior rather than stagnant-lid behavior at high yield stress? If yield stress continued to increase, would stagnant-lid behavior eventually be found? It is also interesting that the squishy lid regimes only show up at moderate internal heating rates, giving way to stagnant lid at higher internal heating rates. One would think that more heat would favor melting and therefore squishy lid behavior, so this is also an area where explanation of the underlying physics would greatly help.

Figure S1 also shows interesting trends that should be interrogated more fully. It is not surprising that lowering the activation energy favors mobile lid behavior. Is this just due to the viscosity contrast between lithosphere and underlying mantle diminishing to the point where hardly any yielding is needed to reach a mobile lid regime? The paper discusses things in terms of coupling between mantle and lithosphere, which is a good explanation of at least part of the physics, but another effect of this is the amount of weakening needed in the lithosphere to initiate subduction. E.g. Solomatov 2004 "Initiation of subduction by small-scale convection" gives a criterion of needing to reduce the viscosity in a lithospheric plate boundary to ~3000 times the underlying mantle viscosity (Wong & Solomatov 2015 "Towards scaling laws for subduction initiation on terrestrial planets: constraints from two-dimensional steady-state convection simulations" then expands on this criterion). So the lower the activation energy the easier it is to meet this criterion.

Detailed comments:

Lines 57-58: Are "ridge only" and "sluggish lid" necessarily always the same? I would argue ridge only is a subset of sluggish lid, but not all sluggish lid behavior results in a ridge only convection planform. For example, the behavior with grain-damage I find in Foley (2018) "The dependence of planetary tectonics on mantle thermal state: applications to early Earth evolution" & Foley (2020) "Timescale of short-term subduction episodicity in convection models with grain damage: Applications to Archean tectonics" I'd argue is sluggish lid, but there is still subduction, so it is not in a ridge only regime.

Lines 69-73: I don't quite buy the arguments against heat pipe regime given here, especially when Io looks like a great example in our own solar system. I see that heat pipes are not shown as a regime in the results, like in Figure 3. Is this because heat pipes just do not develop naturally based on the model assumptions, or are they assumed to not be possible based on the arguments on lines 69-73 and therefore ignored? If it is the former, it would be worth some comment on what about the model setup prevents heat pipe behavior from developing, and then comparing that to what is seen with Io. Could provide some interesting new insight on this type of behavior.

Lines 163-164: This description of the sluggish lid/ridge only regime makes it sound like deformation is widely distributed across the broad convergence/downwelling zone. As a result, it is surprising that plateness is near 1 for this case. I get that the ridge is localized, but isn't there still diffuse deformation on the downwelling/convergence side?

Lines 185-188: Mantle temperature is a bit of a chicken versus egg problem with the plutonic squishy lid, as high temperatures are needed in the first place to generate lots of mantle melt for the intrusions.

Table 1: Not sure it makes sense to give absolute surface velocity cutoffs here, when these could change with different Rayleigh numbers or other factors. Something like mobility, though, that normalizes with interior velocity (and hence overall convective vigor) is more robust.

Line 349: dynamic recrystallization is listed as a healing process, but the term dynamic recrystallization is usually used to

refer to grain size reduction during dislocation creep, so not a healing process.

Line 371-372: The cold stagnant lid end state needs to be justified better. It makes sense planets could end up here, but this isn't shown in the numerical model results, so why planets should go to this state needs better justification.

Line 541: "instantaneously" used twice

Figure S3: It is not easy to compare the high res and low res models. Could the timeseries output (S3a) be plotted to show both the high res and low res models together on the same figure? Then it is easy to compare that they show the same basic dynamics.

Brad Foley

Reviewer #3

(Remarks to the Author)

This work by Lyu et al. addresses the question about what tectonic regimes a terrestrial planet, like the Earth, can exhibit under what conditions, and how does this may change through time. The other uses a state-of-the-art numerical model to explore these points and run a suite of planetary-scale experiments in which they vary some rheological parameters (the surface yield stress, the activation energy of the upper mantle) and thermal conditions (internal heating rate and core-mantle boundary temperature). The model is obviously simplified compared to real planets in many ways, but the authors generally acknowledge and discuss that appropriately in their work. One key finding of the study, is the definition and existence on a previously undetected regime, the Episodic-Squishy Lid, which the authors claim to be possibly relevant for the early Earth and/or modern Venus. Apart from this, the authors compile regime diagrams indicating in which region of the explored parameter space the different regimes may exist. They also propose to propose quantitative measures to distinguish these regimes.

While I enjoyed reading this work, however, I realized that it is "just another" parametric study exploring the a small part of the massive parameter space of terrestrial planet evolution. The search for planetary tectonic regimes with numerical models has been on for decades by now; recent advances in the complexity of such modelling has opened the door for new previously undefined regimes. And one such regime has been found in this study, which is great. However, I was a little disappointed with how this new regime is presented in the paper. Given that it is a transient regime (episodic), I would have interesting and useful to know its dynamics in more detail. What triggers its onset and/or its cessation? How long are such episodes and what determines these time scales? ... These are the questions for a newly detected regime and I wished the authors would have focused their effort more towards this and illustrate the regime using sequences of snapshots and so on to properly introduce their new finding. Instead the authors focus on a few global diagnostics (mobility, plateness, and surface velocity) to map out under which conditions the regime may occur. This is also useful, but the authors themselves acknowledge that this is difficult to do for episodic regimes (e.g. the new regime is excluded from Table 1 where quantitative measures for the different regimes are given) and it never came really clear to me for which ranges of diagnostics the new regime has been assigned by the authors. This may be a crucial detail, because it is not really surprising to find an "episodic squishy lid" in the regime diagram between just between the "episodic lid" and the "squishy lid" and without clear demonstration of its dynamics and diagnostics one may question whether it is really a new regime or just the transition between two previously-known regimes. Despite this main criticism, I think, this work is still interesting and useful for the geodynamics community, and suitable for publication (after some moderate revision).

The study is generally well structured and rather easy to follow. Some typos and language issues persist, but these are easy to fix in later stages of the publication process and did not disturb reading the manuscript. The figures are mostly of good quality, too (but see some comments below). Thus, the quality standards of a journal like Nature Communication are mostly already met or can certainly be met with some minor revisions.

Detailed comments

Abstract:

- Line 18: "planetary dynamo action ..." You never really touch upon this in the manuscript, so it should either be removed or moved towards the back of the list.

- Line 26: Not really clear what "these obstacles" are, please reformulate.

Introduction

- Line 53: Add "currently" or "at present" before "exhibit"

- Line 56: The proposal of this classical episodic lid for Venus is from the early 1990s and somewhat outdated. We still do not know what the regime of Venus is, but it is not just a bunch of global overturn episodes separated by stagnant lid phases

(see e.g. Review by Rolf et al., 2022, Space Science Reviews). Please rewrite to make this clearer.

- Line 93: By now it is not clear what is meant by quasi-steady state. Can this be briefly outlined and then perhaps elaborated further down? Apart from this, it should be explained here already that no planet ever goes into a quasi-steady state, because of secular cooling and so on ...

- Line 138: Write "i=0" for the mantle and i=1 for the continent" to make it clearer. And you should probably add i=2 for primordial material here already.

Results and Discussion

- Line 107: There are so many parameters determining the planetary regime, why did you choose to explore those listed here? For example, you do omit the mantle reference viscosity, which is often considered a key parameter. Please provide some more reasoning here.

- Figure 1: Are the T- and BS-fields really from the same time snapshot??? Looking at the sluggish lid, for example, I see a lot of basalt sinking down the mantle near the "north pole", but there is no cold thermal structure corresponding to this. Instead, there is a cold downwelling near the "South Pole", which gives me the impression that the two halves are rotated by 180° rather than flipped on a vertical axis through the center of the planet. I think I see the same for the Episodic-Squishy lid, so perhaps it is the case for all regimes? ... I find this very irritating, but it may be a personal preference. If you really want to keep it this way, however, you should add a note about this in the caption, to avoid confusion.

- Figure 1: Obviously, you cannot show the dynamics of an episodic regime with a single snapshot. I wished you added a sequence of snapshots for the new ESL regime to understand it better!

- Line 132: "Well-distributed" What does this mean? Do you mean a uniform distribution? In any case, according to Tackley 2000 a value of $P=0$ means that deformation is distributed as in a corresponding isoviscous case, which however still has a distinct deformation pattern (i.e., more deformation around up- and downwelling, less in between), and is not uniform.

- Line 133: "mantle-convection velocity"  velocity of the convecting mantle

- Table 1: Hmm, several studies have done that before and try to squeeze the regimes in diagnostic ranges, but the limiting values remain always diffuse and dependent on several parameters and the subjective interpretation of diagnostics (like how much standard deviation does a regime need to be ESL rather than PSL). It is not clear how you determined for limiting values, like $P=0.5$ or 0.7 . For example, this would highly depend on your definition of P . Some studies use the surface area in which 80% of all surface deformation occur, others have used 90%, and there are other versions yet again. Anyway, you should also include the episodic regimes here, too. If I understand correctly, this should work by adding the standard deviations, right?

- Line 231++: I agree that it make sense to study "hot" and "cold" scenarios, but you should mention already here what this may means in terms of planetary age: hot=young, cold=old. Apart from this, the range of R_h values look fine, but is $TCMB=4250$ K really a good value for a hot/young planet? It seems more like a high-end value for Earth's present CMB temperatures. Also, since you discuss planetary pathways later on, you should have considered a "future" case with R_h and $TCMB$ lower than today. You should already highlight here, that you co-vary them, so there are not independent. This opens up the question, which one is more important though (I am pretty sure it is R_h)!

- Line 233: A rheology with a yield stress is always non-Newtonian, isn't it?

- Line 235: This statement, which you refer to in multiple places in your study, is based on findings from the 1980s with very simple convection models (e.g. no yielding etc.). I do not think it has ever been confirmed for state-of-the-art models like the one used here. There has been some work looking at this (e.g., Arnould et al., 2023, GRL and also Lenardic and co-workers 2019?), but how the inclusion of dislocation creep matters is not really understood yet, I would say. Therefore, I suggest you rephrase your statement here accordingly.

- Line 296++: I was really excited when I saw the title of this section, but rather disappointed after reading it. Frankly, I am not convinced that the numbers you provide here are quite transferrable to other studies (which you already question yourself). Also, how you defined the limited values is not sufficiently clear. Do not get me wrong, Figure 5 still adds value, but unless you want to elaborate on it, I suggest incorporating this short section and the figure into the previous sections.

- Figure 3: Please add to the caption that the lowers panels are just zoom-ins to highlight the small values obtained in the stagnant lid regime and the PSL regime ... I actually wonder if the differences come out better when plotted in log-scale? ... It may be me, but I didn't find it too easy to distinguish between pale orange and pale red in the bottom panels. Please consider a different color combination.

- Line 325: "in many previous studies" A good place to cite a few of those.

- Line 330: Indeed, this is not clear, but you should try to discuss this in more detail as it is one of the key questions. There is

an interesting work by Stern et al. (2018), *Geoscience Frontiers* 9, 103-119, which may be useful for this (see their Fig. 3). In this context, the question about how to transition from one "stable" regime to another is relevant. Going from a mobile/episodic lid to a stagnant lid is typically thought to be easier than going from a stagnant lid back to a mobile lid, because to hysteresis. See e.g. works by Lenardic & Weller, or Weller & Kiefer 2020.

- Line 334: I am not quite sure what this is supposed to say. You mean a quasi-steady state is reached after 1-2 Gyr IN YOUR MODELS? This is highly dependent on the initial conditions and other things.

- Paragraph starting in line 345++: This paragraph adds indeed interesting and relevant discussion. To improve it further, I think the authors should touch upon the interplay of interior and atmosphere evolution (without too much detail, of course): Tectonic activity modulates outgassing activity, which can alter atmospheric composition and eventually change surface temperature, which feeds back into crustal rheology, ... There is work by Gillmann et al. 2014, *JGR* about this for example, even though somewhat specific for Venus ... Another aspect I would like to see addressed in this discussion is that the assumption of one global value for the YS is probably not very realistic (even if changed through time). Deformation can be highly localized and can also occur more likely in specific regions, determined by initial conditions or other processes. This could lead to a spatially heterogeneous yield stress which can trigger very different evolutions than some averaged uniform value. Some examples of causes: specific insolation patterns (Mercury, see Tosi et al. 2015), giant impacts causing hemispheric scale crustal dichotomy (Mars, many works), and for the Earth processes like dehydration stiffening (e.g. Capitanio et al., 2019) and the formation of continents, which add an irreversible crustal structure, likely accompanied with a spatially heterogeneous effective yield stress (e.g. Rolf & Tackley 2011).

- Line 365: I think this could be written in a better way. Here, one can get the impression that it is solely the YS that controls the evolution, but it is actually the contrast between the YS and the stresses YL imposed on the crust and lithosphere, that matter. If $YS \ll YL$, a stagnant lid is expected, if $YS \approx YL$, a mobile regime occurs. In other words, we the imposed stresses increase more than the YS through planetary evolution, the result would still be some mobile regime even though YS may mildly increase. Or, if YL decreases faster than YS through time, the result would be a stagnant lid, even though YS may decrease. While effectively we may arrive at the same point, I think this small difference is still important and should be made clearer in this discussion.

- Line 371: Cite Stern et al. 2018 here (your ref #44)

- Line 396/397: Can we be sure that the intrusion-extrusion ratio was always the same? Perhaps in the early Earth, magmatism was much more extrusive because the crust was initially thinner? I think, it is a bit too early to exclude the heat pipe at least for the Hadean. It seems to be a place where the heat pipe mode may still be active even today ...

- Line 456+: It can help, yes. But you should also mention somewhere that this map explores just a tiny bit of the parameter space, and that there may be more regimes hidden somewhere and multiple ways to transition from another.

Methods

- Line 491/492: I am honestly surprised to about this sentence. Many studies have claimed that mantle viscosity is a very important parameter. After all, it determines the Rayleigh number and thus the thickness of boundary layers, i.e. the lithosphere. In a model, with depth-dependent YS, the lithospheric thickness should be important for the effective strength of the lithosphere. I have a hard time with this statement and I suggest that the authors double-check it, and consider deleting it.

SI Material

- Line 39/40: The values for Rh seem wrong here. Missing factor $1e-12$?

- Line 100++: If you have not tested the resolution for this regime, you should just say so and do not give untested speculations.

Version 1:

Reviewer comments:

Reviewer #1

(Remarks to the Author)

Frankly, I'm not convinced by the authors' replies and the changes they made in the revised version of the MS. Here are the details.

1. I agree that analyzing the evolution of Earth-like planets using a steady-state approach is easier than doing that for transient evolution. However, that does not change the fact that the evolution of the planets is essentially a transient process, and, in my opinion, the authors do not show that their "steady-state" approach is indeed adequate.

2. The authors are excited to introduce a novel tectonic regime for early Earth evolution that combines elements of previously proposed squishy-lid and episodic regimes. Naturally, many transient tectonic regimes might exist between heat

pipe, stagnant, squishy, and mobile-lid regimes. As a result, identifying one that appeared in earlier modeling studies (as authors also agree) but was unnamed is, to my mind, relatively insignificant.

Moreover, I would be careful in taking all these regimes very seriously based on the modeling of the type presented in this MS and other similar models. The reason is that in those models, the plastic strength of the lithosphere is assumed to be constant and homogeneous, a so-called von Mises type of plasticity. That is a strongly simplified approximation, which surely is wrong for the realistic lithosphere, where strength depends on normal stress, i.e., practically on depth (Mohr Coulomb friction). What is even more important is that in reality, the strength is very heterogeneous, being much lower at plate boundaries than inside the plates. While this simplified approximation played an important role in mimicking plate-like behavior, researchers still seem to forget about these models' limitations by seriously discussing the details of tectonic regimes based on such models.

3. In the ms. and in their reply to my comments, the authors paid much attention to their findings concerning scenarios for Earth-like planets moving from non-plate tectonic regimes to the plate tectonic regime. Therefore, I'll also focus on that issue here.

Again, it is well established that decreasing lithospheric yield stress may result in a change from a non-mobile to a mobile (plate tectonics) tectonic regime. Therefore, a statement that the planet will move from a non-mobile regime to plate tectonics if its lithospheric yield stress decreases enough is trivial. It is more valuable to discuss possible mechanisms for how that could happen.

A relevant dynamic scenario was, for instance, suggested by Korenaga (2007, JGR), who proposed that the penetration of water through cracks due to the thermal cracking of oceanic lithosphere have lowered lithospheric strength and enabled plate tectonics already very early in Earth's evolution, just after the magma ocean stage. Sobolev and Brown (2019, ref 8 in the MS) suggested that the lithospheric strength was reduced due to the lubrication effect of continental sediments supplied to the oceans after the rise of continents, which enabled plate tectonics after a kind of hybrid squishy-lid and plume-induced subduction tectonic regime in the Hadean and early Archean.

Those suggestions may be right or wrong, but the suggested mechanisms are specific. On the contrary, the authors suggest that the decrease in strength may be due to the Earth's cooling without pointing to the possible responsible mechanism.

Their only argument is that many potentially relevant rheological mechanisms, such as Peierls creep or dislocation creep, are thermally activated. However, these mechanisms are slowing down with decreasing temperature and, therefore, cannot contribute to lowering the effective lithospheric yield strength.

The authors suggest the paths that an Earth-like planet could follow through the different tectonic mechanisms during its evolution. Again, as in previous studies, the key changing parameter is lithospheric strength. The starting tectonic regime in the sequence of regimes in a decreasing strength path (Fig. 6a) is a stagnant-lid regime. Why that? In Korenaga's model, the regime is close to the mobile-lid from the beginning. In Sobolev and Brown's model, the starting regime is something close to the authors' episodic-squishy-lid regime and not the stagnant-lid regime.

That shows how speculative and controversial the regime's scheme, as suggested by the authors, is.

Finally, the resolution of the presented models is not impressive either. The 256 horizontal cells in (I guess) half annulus model means cell width about 78 km compared to about 20 km cell width in the recent model with the similar modeling technique (Jain et al, 2022, Front. Earth Sci.).

Reviewer #2

(Remarks to the Author)

The majority of my comments and questions have been answered with the revised version of this manuscript. However, a few of my comments on the original submission I don't think were fully answered in the review response or revised manuscript, probably due largely to some miscommunication on exactly what my points of criticism were. I will describe those here below. None of my comments requires extensive new modeling or reworking of the manuscript (I don't think any new modeling is needed at this stage). They really just concern some of the specific wording and claims being made. As a result, I think only minor additional revision is needed before the paper can be accepted.

Remaining comments:

1. The revised manuscript is still claiming that running models to statistical steady-state is solving the problem of hysteresis (e.g. lines 95-102). But just running models to statistical steady-state does not get around this issue. In Weller's work that I cited in the previous review, they documented cases where two models could have the exact same input parameters but end up in different tectonic regimes *at statistical steady-state based* on initial conditions. The different initial conditions are comparing starting from a fully developed mobile lid state to starting from a fully developed stagnant lid state, and Weller & Lenardic found that they needed much lower yield stresses in some cases to get mobile lid convection to result with a stagnant lid initial condition than with a mobile lid initial condition. Changing the initial mantle temperature, as was done in the supplement, is not a full test of the effect of initial conditions in the way Weller & Lenardic did.

I don't think this manuscript needs to do such a full test; I just don't think it should claim running to statistical steady-state avoids issues with hysteresis. The results of this manuscript are still interesting even with hysteresis remaining as a caveat.

2. Lines 65-66 say that the sluggish lid regime is sometimes called the ridge only regime. I don't quite agree because as I said before this implies that sluggish lid and ridge only are interchangeable terms. When in my opinion they are describing different things; I would place ridge only as a subset of sluggish lid, but not all sluggish lid is ridge only.

3. Line 164: I've never heard anyone call episodic lid behavior "hysteresis." Instead, I've heard hysteresis used to describe

what I said in point 1; models that go to different end states due to different initial conditions despite the same parameters.

4. Lines 174-175: I still don't get why the sluggish lid has a very high plateness. The revised text is clearer than the original version, but still doesn't explain it. If the convergence zone is broad, that at least sounds like deformation is distributed across this wide zone, which should make plateness lower. I wonder if there is a figure that could be added (even just in the supplement is fine) helping illustrate how the broad convergence zone still leads to high plateness, when in my mind high plateness requires narrow, localized deformation zones.

5. Line 341: Are small and large grain sizes here flipped? Hot upwellings should lead to larger grain sizes due to high temps and rapid grain growth, with cold downwellings having smaller grains due to sluggish grain growth.

6. Lines 356-368: I have no objections to this paragraph, but it isn't really answering the comment in my first review that sparked this. In my original review I was asking about Fig. S1, where we see that below E_{UM} of about 100 kJ/mol, there is no stagnant lid. It makes sense that at some point E_{UM} is low enough that even with no yielding or other weakening, it still will not produce a stagnant lid. Viscosity has to be about 10,000 times larger at the surface than the mantle interior to produce stagnant lid convection, as shown in Solomatov, 1995; otherwise there will be something like sluggish lid convection. All I was asking here originally is if we were seeing that transition at low E_{UM} in Fig. S1 - do we not see stagnant lid convection at low E_{UM} in Fig. S1 just because the viscosity variation is too low to ever produce a stagnant lid?

7. Line 399: To be pedantic, grain size reduction in the Bercovici damage theory is not driven by stress alone, but by the deformational work, product of stress and strain-rate.

Brad Foley

Reviewer #3

(Remarks to the Author)

The authors have done an excellent job in revising their original draft. I see that all of my comments and concerns have been very satisfactorily addressed. I am particularly please about the fact that the authors have put great effort into improving and/or adding several figures to the revised version, and that they have tried to formalise the method to assign a tectonic regime to each case, based on transparent and quantitative criteria. While details of this method can certainly be discussed and improved in future studies, this approach is a good example for the community about how tectonic regimes should be analysed.

I only have three, very minor comments on the revised version. These can be easily added in the proof-reading stage after acceptance. Otherwise, I will be happy to see this work published soon.

(1) Line 79: „realistic ratios of intrusion to extrusion“: Given that this ratio is quite uncertain and rather debated, it would be good to add one or two references here.

(2) Line 102: „predict the tectonic history of rocky planets“: given the assumptions (like the statistical steady-state), I do not think your models are PREDICTIVE in terms of tectonic history yet, but rather SUGGESTIVE. Consider rephrasing here.

(3) Line 174: You seem to have added a phrase about double-sided subduction here. I do not think this is necessary at all, but if you add it, you should also add a half-sentence that „this may not be representative of present-day subduction on Earth and is probably at least partially caused by our relatively simple rheological model“.

[editorial note: reviewer name redacted]

Reviewer #4

(Remarks to the Author)

I've read the paper and the first set of reviews, and also the 2nd review from Referee 1.

Admittedly I'm not a giant fan of such parameter sweeping studies that are sort of a random fishing expedition posing as hypothesis testing. But there seems to be at least more positive response from the other referees. While I mostly agree with Ref 1 that his/her comments were not adequately addressed, I think the authors could address the concerns with a little more thought.

1. Using statistically steady state solutions to elucidate evolution and transitions between states has limitations. It can tell us something about the state on either side of a transition but very little about the transition itself (how long it takes for example). Arguing that it's easier to do steady state is not a good rebuttal. But arguing that bracketing transitions is useful (i.e., articulating where you are to where you end up is just as important on a roadmap as an estimate of how long it will take you to get there). So I think there is still utility, but the authors should make a better case.

2 and 3 Combined. First, yes I agree that finding a new regime that isn't really new, just not named yet, is not terribly novel. But putting into the context of other regimes and transitions, is not useless. But more to the point about the treatment of lithospheric yield stress: To be up front, I believe the pervasively used yield stress approach is fundamentally flawed. The yield stresses are always some unrealistically small value. But more important (and as I and others like Mike Gurnis and most recently Fuchs and Becker have written about for 30 years) it fails a fundamental test. That is, a yield stress allows weakness to occur suggestive of a plate boundary, but as soon as the stress and deformation cease, the weak zone vanishes, which is not physical or even geological. Dormancy and inheritance of weak zones is well known and important for plate boundary activation and reactivation; and there are plenty of extant physical models to explain this (yes, grain-damage, but also others involving state variables like temperature, water content, anisotropy). The yield stress formulation is widely used because it's easy and it is what is made available in large codes (STAG and CITCOM), which many folks use and just more or less ignore all the arguments about why it is a faulty approach. Having gotten that off my chest, the current study is what is and perhaps, since it employs a steady-state approach, it can be argued that it side-steps the issue of weak-zone dormancy/inheritance which are strong temporal effects. And, just to try to be constructive and helpful, perhaps these effects might be used to answer some of the Referee's criticisms. First, I do agree with the Referee that arguing the transition from stagnant to mobile with a decrease in yield stress is trivial; it is indeed obvious but that's not necessarily fatal. But the reason why it decreases is indeed not explained in the paper and the Referee is justified in wanting the authors to providing some physical explanation about why the yield stress decreases with time. If the authors can't or aren't willing to address that, then I would consider that fatal. But I will throw a possible lifeline, or a few of them. There are a few physical mechanisms that might help them resolve this, and in fact the suggestions have been around for pushing 20 years already. It is true (as the Referee says) that the effect of cooling would at face value have a stiffening effect; and that has been known for a long time. It's the same problem of why a planet with a cool surface like Earth has mobile lid (i.e., plates), while one with a hot surface like Venus does not. But that was the entire point of grain damage theories, going back to the papers of Landuyt and myself. and elaborated on with multiple papers by myself and others since then. That is, healing becomes less efficient with cooling, while the damage processes become more efficient; so the two both conspire to make damage (and lasting, dormant damage) prevalent on a cooling planet, but less so on a hot planet. That is one way the so-called or effective yield stress might decrease with time. But part of the same effect, or any effect that builds and accumulates dormant weak zones (not just mylonitization but fractures, hydration, anisotropic fabric), is that old unhealed scars progressively make the lithosphere weaker (Gurnis and Zhong would have made this case in the mid-1990s). Finally, Karato and Barbot (2018) offered a mechanism to imply that dynamic weakening is temperature dependent as a mechanism to explain the Earth-Venus dichotomy also. It's not that one of these is right and the others are wrong; they are all probably activated in some way. But the point is, if the authors took some of these microstructural and rock mechanics effects into account and didn't just rely on a faulty yield-stress formalism because so many others have (plurality is not the same as validity), they could probably strengthen their case.

On the numerical resolution comment. If it's just about being impressive, I don't see why that matters. The question isn't about how big your grid is, but whether it does the job and that the solutions are well resolved.

[editorial note: reviewer name redacted]

Version 2:

Reviewer comments:

Reviewer #2

(Remarks to the Author)

The revised manuscript has satisfactorily answered all of my comments from the previous review. However, I hesitate to say that I think the paper is ready for publication as in my opinion the discussion section is starting to overstep the bounds of what this paper can really say based on the modeling performed.

I view the contributions of this paper being the way metrics (mobility, plateness, etc.) were analyzed to assign regimes in a novel way and the final regime diagram. These are both solid contributions to geodynamics. The regime diagram can then form the basis for some speculation about evolutionary paths Earth or Venus might have taken. And my view was that previous versions of the manuscript did this while still correctly conveying that this was a speculative interpretation of the results; different models with a different design and philosophy would be needed to make a strong argument for any particular path or to show that one path better satisfies observations over another path. In my opinion that is outside the scope of this present manuscript and better left for some future paper with a different modeling design to address.

In response to other reviewer comments, though, I think the manuscript has gone overboard trying to argue for a particular evolution for Earth (and to some extent Venus), beyond what the results can support. I see two main problematic aspects of the current manuscript which stem from trying to argue specifically for the top right to bottom left path through Figure 5b.

1. The manuscript now heavily appeals to physical processes like grain size evolution or thermal effects on earthquake rupture that are not included in their models to justify this path. The argument isn't very convincing since we don't know if the regimes would show up in the same way and in the same parameter space if the models actually did include these other effects.

2. The manuscript also tries to appeal to observations to justify their preferred evolutionary path for the Earth. But despite stating that the manuscript will “discuss the geological evidence” “for decreasing lithospheric strengths over time” no actual observations of the early Earth are given. Instead, the manuscript discusses models (both conceptual cartoon models and physical models) that other studies have put forth. While the models in those cited papers are themselves based on observations (for the most part), they are particular interpretations of the observations. Appealing to other peoples’ interpretations rather than directly to observations themselves to support the argued for evolutionary scenario is unconvincing. Along the same lines there are repeated claims about how the regimes and evolutionary path the paper argues for can reconcile the geologic record, yet no specific observations are outlined nor is there any demonstration that the models can explain any key observations about the early Earth.

My opinion is that these shortcomings stem from trying to argue beyond what the key results can really support. While one approach could be more models and analysis to firm up these unconvincing arguments, I don’t think that is feasible for this paper as a wholly different modeling philosophy and design would be required to, e.g., really show that the models do explain key early Earth observations or to include effects beyond just plastic yielding that might lead to effectively weakening lithosphere over time. Such an exercise is best saved for a future paper.

The authors need to reassess what they can justifiably argue based on their results and keep the discussion focused on this. There are still interesting things that could be said about possible evolutionary scenarios and observables that might distinguish between them that could be explored in the discussion while staying grounded in the results presented. But arguing for one particular path is a heavy lift, as the rounds of review are showing, since the model design was not set up to do this.

Brad Foley

Reviewer #4

(Remarks to the Author)

The authors have done a commendable job of responding to multiple referee comments, including my own, with clarifications and additional work and information. I see no reason not to proceed.

Version 3:

Reviewer comments:

Reviewer #2

(Remarks to the Author)

As the authors mentioned in their cover letter to the editor, they discussed the changes they made with me before resubmission. They provided good clarity on their logic of how they took the model results and applied them to consider possible evolutionary scenarios for the Earth and Venus. As a result all my comments from previous reviews have been addressed and I recommend the paper be accepted.

Dissecting the puzzle of tectonic lid regimes in terrestrial planets

Tianyang Lyu, et al.

(Response to the reviewers' comments)

REVIEWER COMMENTS

Reviewer #1 (Remarks to the Author):

Comment: The ms. reports results of the comprehensive modeling study of the long-term thermo-mechanical evolution of planet with Earth-like parameters. The study uses a similar modeling approach and continues the previous modeling studies by the ETH team (i.e. Lourenço et al., 2018, 2020). New is that the authors investigate effect of different model parameters than it was done in the previous studies and use “quasi-steady state” approximation. In particular they assume that the CMB temperature and internal heat production is fixed and run model till 10Gy to approach “quasi-steady state”. This allows to attribute different tectonic styles to the domains in two parameter space, where internal heat production/CMB temperature is one parameter and effective lithospheric stress (cohesion) another parameter. In addition, they also investigate effect of changing of activation energy in the upper mantle on tectonic regime. I think that all these models are interesting as some theoretical end members, but I doubt that they can be applied to the evolution of the realistic planets. It is well known that internal heat production in planets strongly change during the evolution due to the decay of radiogenic elements and due to their redistribution from mantle to the crust. CMB temperature also is significantly decreasing. Authors claim (lines 333-335) that their models are applicable to transient planetary evolution with reference to their supplement. However, I did not find convincing evidence for that in their supplement information (see also below).

Reply: We fully understand and agree with the point of the reviewer that the internal heat production and core-mantle boundary (CMB) temperature of terrestrial planets gradually decrease over time. Indeed, these parameters evolve significantly, particularly during the early stages of planetary development, and their decline has a profound impact on the mantle's thermal structure and dynamics.

However, as pointed out by the reviewer, employing a gradually cooling model (such as core-cooling or the decay of radiogenic elements) to account for such changes, while being more realistic, presents several challenges. Most importantly, it is difficult to clearly identify the characteristic regime for given physical conditions, because the

thermal condition changes on a timescale that is of the same order as the typical length of episodes in various episodic regimes (0.5 to ~2 billion years).

A weakness of the transient approach is exemplified in Lourenço et al. (2016), where a large part of the parameter space is labelled as “potentially unstable”. In this parameter space, episodic behavior can occur, but usually does not occur in every model run (i.e., due to hysteresis). As models are re-run for the same parameters, either episodic or stagnant behavior occurs in this parameter space.

To address these challenges, we choose a simplified approach. We adopt a statistical-steady state approach, maintaining constant CMB temperature and internal heat production over extended time scales. This method enables us to gather long time series of predicted diagnostic criteria such as mobility and plateness. Then, we analyze these time series quantitatively to classify the tectonic regime (Tables 1-2, Figs. 4, 5a). Over long timescales, episodic regimes can be better captured and analyzed. Note that we now take advantage of this dataset more extensively, significantly improving the classification of tectonic regimes compared to the initial submission (Tables 1-2, Fig. 4).

For fixed thermal conditions, an (idealized) planet will evolve into a given regime at least within 1-2 Gyrs. This is the typical timescale it takes our models to reach the statistical steady-state regime from the initial condition. Except during early evolution, the thermal conditions of a planet do not change much over this timescale. That being said, we still expect our models with stagnant lid and plutonic-squishy lid to be applicable to the early evolution of terrestrial planets, because they exhibit high internal temperatures even in the statistical steady state. Thus, our simplified approach allows us to discuss regime shifts during the evolution of an idealized planet following a planet's cooling trajectory in the regime diagram (i.e., moving downward over time).

To address this comment, we better explain our approach and its limitations near the beginning of the section “Relevance of lithospheric yield stress for planetary evolution”. We now also show that the preferred tectonic regime in each case is independent on the initial geotherm (see our response to Reviewer #3 and the accompanying supplementary figure). We also highlight that our models typically reach the same regime as in the steady state early in model evolution (lines 376-380).

Comment: What is likely robust in the models in the ms. is that at the low cohesion the tectonic regime is always mobile with fast moving surface, at high cohesion the regime is low-mobility with low surface velocities and the intermediate mobility regimes correspond to the intermediate values of cohesion. However, this qualitative result is not new and well described in previous studies (e.g. Lourenço et al., 2018, 2020).

Reply: As mentioned in the MS, the steady-state approach is new, a simplification we

view as a strength, because it allows us to determine regimes more quantitatively, and actually even to establish classification criteria. Therefore, there are many other (arguably more novel) results in our study which are robust. While Lourenço et al. (2016, 2020) established foundational classifications of tectonic regimes, our research introduces additional modes, specifically the Sluggish Lid and the newly identified Episodic-Squishy lid (ESL). These modes capture tectonic behaviors not previously addressed. We also put all tectonic regimes in context to each other covering a parameter space that can be understood as the thermal evolution of the planet (vertical axis of Fig. 5) vs the effective yield stress of the lithosphere. Finally, we also expand (notably in the resubmitted version) the definitions and classification criteria of tectonic regime, an effort our community will benefit from as more tectonic regimes are discovered.

Comment: The quantitative results of the models in the ms. are unlikely applicable for the planets in the realistic transient regime. For instance, their results do not match with the modeling results by Lourenço et al. (G3 2020), obtained with the same parameters, but for the transient Earth. Compare Fig.3a from the ms. with Fig.3a from Lourenço et al. (2020). At eruption efficiency of 10% average mobility value above 1 in the transient Earth model by Lourenço et al. (2020) is achieved at cohesion lower than 30 MPa although in the ms. that happened at cohesion lower than 100 MPa. That means that the detailed diagrams of tectonic regimes presented in ms. are unlikely qualitatively applicable to the transient planetary evolution.

Reply: The reviewer is correct in pointing out that there are some differences between our model predictions and those of Lourenço et al. (2020). However, this does not imply that our results are wrong, or that these differences are mostly due to our statistical-steady state approach. We remind the reviewer that the extent of changes in internal heating and in CMB temperature are small over the last 2-3 billion years (i.e., in Lourenço et al., and in real planets in our solar system). There are other differences in the setup between Lourenço et al. (2020) and our study, notably in terms of the partition coefficient of HPE during melting, which explain these differences better. The partitioning coefficient of HPE during mantle melting is 0.01 in our study but was 1.0 in Lourenço et al. (2020). While 0.01 is a much more realistic value (HPE are incompatible), this difference strongly affects internal heating and thus the viscosity of the crust with implications for tectonics. This effect has been explicitly studied in Lourenço et al (2018). We now clarify this assumption and apologize to the reviewer for missing to report it properly in the previous version of the MS.

That all being said, we are unsure if the differences between the results of Lourenço et al. (2020) and our study are actually large. Unfortunately, Lourenço et al. (2020) do not plot time series of mobility and plateness. But looking at the predicted surface velocities of their cases with low extrusion efficiency (10%) in their Figure 4, we infer that their cases with **100 MPa < Surface Yield < 260 MPa** overlap with our definition of the

episodic-squishy lid (ESL). That being said, even though they may have had these cases in their dataset, they have not recognized their relevance in terms of presenting a distinct regime that exhibits periods of high mobility. This again underscores the importance of quantitative analysis of long time series in the statistical steady state.

To address this comment, we now clarify and justify our assumptions in terms of HPE partitioning. We also mention that cases that likely match our criteria for ESL have been predicted by Lourenço et al. (2020) in a similar parameter range (lines 329-331).

In the discussion the authors suggest evolution scenarios for the planets in the cases of lithospheric yield stress increasing or decreasing with time. However, these general scenarios are obvious and could be suggested based on previous studies.

Reply: We are not aware that these scenarios actually have been suggested already, even based on previous studies. In other words: we agree that this is low-hanging fruit, but we are not aware of anyone having picked it in this context.

While the concept of planetary evolution based on lithospheric yield stress changes is grounded in established research, it remains a fundamental aspect that cannot be overlooked in the context of our study. Importantly, even though a detailed review of existing modelling results may have been able to reach similar conclusions in terms of how a planet evolves during cooling, our study for the first time (to our knowledge) includes a detailed discussion of how changes in effective yield stress during planetary cooling may affect tectonic evolution. In particular in the context of newly-found regimes such as the plutonic-squishy lid and episodic squishy lid (and sluggish lid), and not just in the context of changes between the stagnant lid and plate tectonics. In addition, our discussion in this regard is based on a series of self-consistent models (all using the same basic setup). Putting all six regimes on a single regime map, and discussing planetary tectonic evolution in a newly-informed way accordingly, is one of the novel aspects of our study.

Reviewer #2 (Remarks to the Author):

This manuscript presents a large set of numerical convection models, mapping out regimes of global tectonic behavior as a function of yield stress and internal heat production plus core temperature, as well as viscosity activation energy shown in the supplement. The main purpose is to develop quantitative criteria for classifying models into these regimes and show how the regimes are affected by these key model parameters. The paper identifies a new regime, the episodic squishy lid regime, and discusses how planets might evolve over time through the space of different tectonic regimes mapped out. A critical uncertainty is how the yield stress itself might change through a planet's evolution, dictating the path the planet then takes as it cools.

The paper is well written, provides new insights on mantle convection regimes, and the conclusions are for the most part supported by models presented. There are a few important points that need to be further developed or considered better, but these can be handled with a revision. I think the paper will be suitable for publication after revision.

Main Comments:

1. The paper states on lines 105-107 that issues related to hysteresis and initial conditions are resolved by running the models for ~10 Gyrs to reach quasi-steady-state. While I think the approach of analyzing models in quasi-steady-state is a good one, this does not eliminate issues with hysteresis and initial conditions. In Weller & Lenardic's work, they ran models starting from different initial regimes (stagnant lid versus mobile lid) to quasi-steady-state and still found that in some cases the initial condition dictated the final state. Models with the exact same parameters could end up in different regimes, even when quasi-steady-state is reached, based on starting in different initial convection regimes. So just running models for a long time does not resolve the issues with hysteresis. I don't think this paper necessarily needs to fully explore hysteresis the way Weller & Lenardic did. But it should at least clearly report the initial conditions for context here. Are the models initially started from a mobile lid or stagnant lid state? What do the initial temperature, viscosity, and velocity fields look like? These are important for then interpreting the quasi-steady-state regimes that develop.

Reply: We now clearly report the initial conditions in the Method section, and show the initial T and viscosity fields in a Suppl Fig. We clarify that we do not impose any initial velocities; hence a figure showing the initial velocities (being zero everywhere) is redundant.

2. The paper doesn't give much physical insight into why the regimes develop in the parameter space shown. I think more should be done here, as the regime diagrams are the key result of the paper. While previous papers have developed scaling relationships for, e.g., convective stress and applied these to explain boundaries between mobile and

stagnant-lid regimes, this manuscript has a more comprehensive set of regimes mapped out, and therefore providing physical insight into why the regimes lay where they do would make for a more significant contribution.

Reply: This is a very good suggestion. We have expanded our manuscript by adding a paragraph (lines 329-334) that compares our results with previous studies, such as those by Lourenço et al. (2020), and clarifies the similarities and distinctions in the identified tectonic regimes. Additionally, we have included two paragraphs (lines 347-368) that discuss the underlying mechanisms, focusing on the effects of plate-mantle coupling and the influence of lithospheric rheological heterogeneity, particularly due to magmatism, on tectonic style. These sections explain how variations in E_{UM} modify the viscosity contrast between the lithosphere and asthenosphere, thereby controlling the transition between mobile and stagnant-lid regimes, and how magmatism introduces rheological heterogeneity that facilitates localized yielding and the development of different tectonic regimes under various yield stress conditions.

For example, in Figure 3b, it is not clear to me why stagnant-lid behavior is only seen at high internal heat rates, rather than being a regime that can always be reached at high yield stresses.

Reply: Thank you for this important point. We have added one paragraph (lines 347-355) to address this point in more detail:

“These models with variable E_{UM} help to understand the behavior of our main model suite in Fig. 5. Stagnant lid cases at high thermal conditions are promoted by less efficient plate-mantle coupling (hot asthenosphere). With decreasing mantle temperatures (Fig. S2a), plate-mantle coupling increases, promoting the plutonic-squishy lid (with intermittent yielding), and then the sluggish lid (with dominant mantle drag and coherent tectonics) even at the highest yield stresses explored here. In turn, for low yield stresses, mobile-lid behavior (with highly-active tectonics) occurs. The transitional episodic and episodic-squishy lid regimes naturally occur in the parameter spaces in-between (Fig. 5b).”

Why do low internal heating rates result in sluggish lid behavior rather than stagnant-lid behavior at high yield stress? If yield stress continued to increase, would stagnant-lid behavior eventually be found?

Reply: Good point! Indeed, we expect that the sluggish lid will ultimately transit into a “cold” stagnant lid, because of a much thicker lithospheric boundary layer and a strongly reduced mantle Rayleigh number. We make this point in the paper when we discuss Fig. 6 (lines 426-427).

It is also interesting that the squishy lid regimes only show up at moderate internal heating rates, giving way to stagnant lid at higher internal heating rates. One would think that more heat would favor melting and therefore squishy lid behavior, so this is also an area where explanation of the underlying physics would greatly help.

Reply: We reply to it above. Now discussed in the second but last paragraph of the section “Parameter Study”.

Figure S1 also shows interesting trends that should be interrogated more fully. It is not surprising that lowering the activation energy favors mobile lid behavior. Is this just due to the viscosity contrast between lithosphere and underlying mantle diminishing to the point where hardly any yielding is needed to reach a mobile lid regime? The paper discusses things in terms of coupling between mantle and lithosphere, which is a good explanation of at least part of the physics, but another effect of this is the amount of weakening needed in the lithosphere to initiate subduction. E.g. Solomatov 2004 "Initiation of subduction by small-scale convection" gives a criterion of needing to reduce the viscosity in a lithospheric plate boundary to ~ 3000 times the underlying mantle viscosity (Wong & Solomatov 2015 "Towards scaling laws for subduction initiation on terrestrial planets: constraints from two-dimensional steady-state convection simulations" then expands on this criterion). So the lower the activation energy the easier it is to meet this criterion.

Reply: In our study, rheological heterogeneity in the lithosphere is mostly provided by intrusive magmatism. Now discussed in the last paragraph of the section “Parameter Study” (lines 352-364):

“In addition, lithospheric structure controls the tectonic regime. Large lateral rheological heterogeneity is required to induce localization of deformation and yielding. In our models, localized weakening is predominantly provided by intrusive magmatism. Hot stagnant-lid cases exhibit a thick lithosphere with only moderate magmatism, partly because massive basaltic melt has already been extracted from the mantle early in their evolution. In contrast, warm plutonic-squishy cases display more magmatism (Fig. S2a) and thus a rheologically heterogeneous lithosphere with frequent active (albeit intermittent) yielding events. Cool sluggish-lid cases again exhibit less magmatism, leading to only passive yielding, typically confined to a single subduction zone globally. Even though only minor magmatism occurs in mobile-lid cases, the plate-mantle system with active subduction self-sustains rheological heterogeneity (e.g., thin lithosphere near ridges) and transfers large stresses distally, sufficient to maintain subduction at low yield stresses.”

Detailed comments:

Lines 57-58: Are “ridge only” and “sluggish lid” necessarily always the same? I would argue ridge only is a subset of sluggish lid, but not all sluggish lid behavior results in a ridge only convection planform. For example, the behavior with grain-damage I find in Foley (2018) "The dependence of planetary tectonics on mantle thermal state: applications to early Earth evolution" & Foley (2020) "Timescale of short-term subduction episodicity in convection models with grain damage: Applications to Archean tectonics" I'd argue is sluggish lid, but there is still subduction, so it is not in a ridge only regime.

Reply: Agreed. We have fixed the description to clarify this. In our sluggish lid, we have ridges and broad two-sided subduction zones. So it is technically not “ridge-only”.

Lines 69-73: I don't quite buy the arguments against heat pipe regime given here, especially when Io looks like a great example in our own solar system. I see that heat pipes are not shown as a regime in the results, like in Figure 3. Is this because heat pipes just do not develop naturally based on the model assumptions, or are they assumed to not be possible based on the arguments on lines 69-73 and therefore ignored? If it is the former, it would be worth some comment on what about the model setup prevents heat pipe behavior from developing, and then comparing that to what is seen with Io. Could provide some interesting new insight on this type of behavior.

Reply: We have toned down our arguments against the heat-pipe mode in the introduction section. We have also added some discussion in lines 331-334 why the heat pipe does not occur in our models (high intrusion:extrusion ratio), and under which conditions it may occur in nature. We have now emphasized in the main text (beginning of results section) that we consider a high intrusion:extrusion ratio (one of our important model ingredients).

Lines 163-164: This description of the sluggish lid/ridge only regime makes it sound like deformation is widely distributed across the broad convergence/downwelling zone. As a result, it is surprising that plateness is near 1 for this case. I get that the ridge is localized, but isn't there still diffuse deformation on the downwelling/convergence side?

Reply: We are sorry that our previous explanation was misleading. We have now reworded this for clarity. In the sluggish-lid models, deformation is sufficiently localized to result in $P=1$. Downwellings are expressed as wide double-sided subduction zones with yielding along a “V-shaped” plate-scale fault system.

Lines 185-188: Mantle temperature is a bit of a chicken versus egg problem with the plutonic squishy lid, as high temperatures are needed in the first place to generate lots of mantle melt for the intrusions.

Reply: The planet starts hot, promoting the plutonic squishy lid. Due to inefficient subduction, the mantle remains rather hot. There is no chicken and egg problem, as the planet starts hot.

Table 1: Not sure it makes sense to give absolute surface velocity cutoffs here, when these could change with different Rayleigh numbers or other factors. Something like mobility, though, that normalizes with interior velocity (and hence overall convective vigor) is more robust.

Reply: We agree and removed these cutoffs. We further improved our regime classification scheme (Fig.4 and Table 1-2).

Line 349: dynamic recrystallization is listed as a healing process, but the term dynamic recrystallization is usually used to refer to grain size reduction during dislocation creep, so not a healing process.

Reply: We removed the reference to dynamic recrystallization here.

Line 371-372: The cold stagnant lid end state needs to be justified better. It makes sense planets could end up here, but this isn't shown in the numerical model results, so why planets should go to this state needs better justification.

Reply: We clarify that the Rayleigh Number would become very low, and we cite a paper that low-Ra planets assume a stagnant lid. We now add that a thicker lithosphere is more difficult to break.

Line 541: "instantaneously" used twice

Reply: Corrected.

Figure S3: It is not easy to compare the high res and low res models. Could the timeseries output (S3a) be plotted to show both the high res and low res models together on the same figure? Then it is easy to compare that they show the same basic dynamics.

Reply: Thank you for your valuable feedback regarding Fig. S3. As the models exhibit highly chaotic behavior, directly plotting the time series of the high-resolution and low-resolution models together would not provide meaningful insights, as their trajectories diverge rapidly despite capturing similar overall dynamics.

To address this, we have adopted a statistical comparison approach. Specifically, we computed and compared the averages and standard deviations of key metrics, such as mobility and flatness, between the high-resolution and low-resolution models, and determined the lid modes based on the time fraction spent in each base regimes (as in Table 1-2). The results of this analysis are presented in Tables S4 and S5 in the supplementary material. These tables provide a clear quantitative measure of the agreement between the high- and low-resolution models over time.

Brad Foley

Reviewer #3 (Remarks to the Author):

This work by Lyu et al. addresses the question about what tectonic regimes a terrestrial planet, like the Earth, can exhibit under what conditions, and how does this may change through time. The other uses a state-of-the-art numerical model to explore these points and run a suite of planetary-scale experiments in which they vary some rheological parameters (the surface yield stress, the activation energy of the upper mantle) and thermal conditions (internal heating rate and core-mantle boundary temperature). The model is obviously simplified compared to real planets in many ways, but the authors generally acknowledge and discuss that appropriately in their work. One key finding of the study, is the definition and existence on a previously undetected regime, the Episodic-Squishy Lid, which the authors claim to be possibly relevant for the early Earth and/or modern Venus. Apart from this, the authors compile regime diagrams indicating in which region of the explored parameter space the different regimes may exist. They also propose to propose quantitative measures to distinguish these regimes.

While I enjoyed reading this work, however, I realized that it is “just another” parametric study exploring the a small part of the massive parameter space of terrestrial planet evolution. The search for planetary tectonic regimes with numerical models has been on for decades by now; recent advances in the complexity of such modelling has opened the door for new previously undefined regimes. And one such regime has been found in this study, which is great.

Reply: We have improved the justification for why we focus on the parameters we chose (lines 113-115). We have also significantly improved the classification scheme to quantitatively distinguish between the six tectonic regimes, significantly beyond what has been done in previous studies (see below; Fig. 4; Table 1-2). We expect that this or a similar classification scheme will be useful for upcoming studies, to ultimately move ahead the search for planetary tectonic regimes with numerical models.

However, I was a little disappointed with how this new regime is presented in the paper. Given that it is a transient regime (episodic), I would have interesting and useful to know its dynamics in more detail. What triggers its onset and/or its cessation? How long are such episodes and what determines these time scales? ... These are the questions for a newly detected regime and I wished the authors would have focused their effort more towards this and illustrate the regime using sequences of snapshots and so on to properly introduce their new finding. Instead the authors focus on a few global diagnostics (mobility, plateness, and surface velocity) to map out under which conditions the regime may occur. This is also useful, but the authors themselves acknowledge that this is difficult to do for episodic regimes (e.g. the new regime is excluded from Table 1 where quantitative measures for the different regimes are given) and it never came really clear to me for which ranges of diagnostics the new regime has been assigned by the authors. This may be a crucial detail, because it is not really

surprising to find an “episodic squishy lid” in the regime diagram between just between the “episodic lid” and the “squishy lid” and without clear demonstration of its dynamics and diagnostics one may question whether it is really a new regime or just the transition between two previously-known regimes. Despite this main criticism, I think, this work is still interesting and useful for the geodynamics community, and suitable for publication (after some moderate revision).

The study is generally well structured and rather easy to follow. Some typos and language issues persist, but these are easy to fix in later stages of the publication process and did not disturb reading the manuscript. The figures are mostly of good quality, too (but see some comments below). Thus, the quality standards of a journal like Nature Communication are mostly already met or can certainly be met with some minor revisions.

Reply: Thank you for the constructive comments! We took these comments seriously and added a detailed description of how the episodic squishy lid works (and added a new Fig. 3 of its time evolution). We also devised a novel and more quantitative classification scheme to distinguish between the six tectonic regimes found here (e.g., Fig. 4; Table 1-2). In a nutshell, this new tectonic classification scheme is based on the fraction of model time spent with (1) high M, (2) low M/high P, (3) low M/low P. Depending on the time fraction spent with these mobilities (M) and platenesses (P), each model case is assigned the relevant tectonic regime. Such a classification can robustly and quantitatively distinguish between all 6 regimes, even though detailed visual analysis is still needed to distinguish between the episodic and episodic-squishy lid. Such an approach is novel, and we hope that it will be applied in future modelling studies.

Detailed comments

Abstract:

- Line 18: “planetary dynamo action ...” You never really touch upon this in the manuscript, so it should either be removed or moved towards the back of the list.

Reply: Fixed.

- Line 26: Not really clear what “these obstacles” are, please reformulate.

Reply: Fixed.

Introduction

- Line 53: Add “currently” or “at present” before “exhibit”

Reply: Fixed.

- Line 56: The proposal of this classical episodic lid for Venus is from the early 1990s and somewhat outdated. We still do not know what the regime of Venus is, but it is not just a bunch of global overturn episodes separated by stagnant lid phases (see e.g. Review by Rolf et al., 2022, Space Science Reviews). Please rewrite to make this clearer.

Reply: Agreed. Corrected.

- Line 93: By now it is not clear what is meant by quasi-steady state. Can this be briefly outlined and then perhaps elaborated further down? Apart from this, it should be explained here already that no planet ever goes into a quasi-steady state, because of secular cooling and so on ...

Reply: Thank you for your suggestion. We have revised the definition (lines 99-101) to briefly outline the concept of the statistical-steady state, and a more detailed explanation is now provided later in the main text (line 373-380). We emphasize that (except early in their evolution), secular cooling of terrestrial planets is rather slow (Andrault et al., 2016). Likewise, the rate of change of internal heating also becomes slower and slower with time.

- Line 138: Write “ $i=0$ ” for the mantle and $i=1$ for the continent” to make it clearer. And you should probably add $i=2$ for primordial material here already.

Reply: Our models do neither include continents, nor do they include primordial

material. To be honest, we do not understand what the reviewer is referring to. Line 138 in the original submission is unrelated to this reviewer comment.

Results and Discussion

- Line 107: There are so many parameters determining the planetary regime, why did you choose to explore those listed here? For example, you do omit the mantle reference viscosity, which is often considered a key parameter. Please provide some more reasoning here.

Reply: Thank you for your insightful comment regarding our parameter selection and the omission of mantle reference viscosity in our exploration.

We chose to focus on parameters such as core-mantle boundary (CMB) temperature, internal heating rate, upper-mantle activation energy (E_{UM}), and effective yield stress because they directly influence the thermal structure, rheological properties, and tectonic dynamics of planetary mantles, thereby governing transitions between tectonic regimes. As noted in the revised manuscript, these parameters are central to controlling the interplay between mantle convection, lithospheric strength, and planetary cooling rates, which are key to understanding planetary tectonic evolution. Additionally, these choices reflect the parameters' ability to map out distinct tectonic regimes systematically and identify new transitional regimes, such as the episodic-squishy lid.

Regarding the omission of mantle reference viscosity, we acknowledge its importance in rheological studies, particularly for determining the overall dynamics of mantle convection. However, as described in Text S1 of the Supplementary Material, the activation energy of the upper mantle (E_{UM}) effectively captures the first-order effects of mantle viscosity variations, especially at lithospheric scales. E_{UM} plays a pivotal role in influencing the viscosity contrast between the lithosphere and the asthenosphere, thereby modulating plate-mantle coupling and tectonic behavior. High E_{UM} values promote stagnant lid regimes by increasing the viscosity of the lithosphere, while low E_{UM} values enhance lithosphere-asthenosphere coupling and favor mobile lid behavior. In Lourenço et al. (2020), no large differences are found between the regime maps with different reference mantle viscosity.

- Figure 1: Are the T- and BS-fields really from the same time snapshot??? Looking at the sluggish lid, for example, I see a lot of basalt sinking down the mantle near the "north pole", but there is no cold thermal structure corresponding to this. Instead, there

is a cold downwelling near the "South Pole", which gives me the impression that the two halves are rotated by 180° rather than flipped on a vertical axis through the center of the planet. I think I see the same for the Episodic-Squishy lid, so perhaps it is the case for all regimes? ... I find this very irritating, but it may be a personal preference. If you really want to keep it this way, however, you should add a note about this in the caption, to avoid confusion.

Reply: Thank you very much for your insightful suggestion. I have made the adjustment as you recommended and now the T- and BS-fields are flipped around the vertical axis through the center of the planet, instead of being rotated by 180 degrees.

- Figure 1: Obviously, you cannot show the dynamics of an episodic regime with a single snapshot. I wished you added a sequence of snapshots for the new ESL regime to understand it better!

Reply: Thank you for your valuable suggestion. We have added a new Figure in the main text (Fig. 3) to illustrate the time evolution of the new ESL regime in more detail.

- Line 132: "Well-distributed" What does this mean? Do you mean a uniform distribution? In any case, according to Tackley 2000 a value of $P=0$ means that deformation is distributed as in a corresponding isoviscous case, which however still has a distinct deformation pattern (i.e., more deformation around up- and downwelling, less in between), and is not uniform.

Reply: Thank you for your comment. We agree with your observation and have revised the text. The term "well-distributed surface deformation" has been amended by a reference to "isoviscous cases" to better reflect the deformation pattern described by Tackley (2000). The more distributed/widespread deformation is the lower P . The lowest value P can attain is 0.0 , which is defined in such a way that it is attained for isoviscous convection.

- Line 133: "mantle-convection velocity"  velocity of the convecting mantle

Reply: Fixed.

- Table 1: Hmm, several studies have done that before and try to squeeze the regimes in diagnostic ranges, but the limiting values remain always diffuse and dependent on several parameters and the subjective interpretation of diagnostics (like how much standard deviation does a regime need to be ESL rather than PSL). It is not clear how you determined for limiting values, like $P=0.5$ or 0.7 . For example, this would highly depend on your definition of P . Some studies use the surface area in which 80% of all surface deformation occur, others have used 90%, and there are other versions yet again. Anyway, you should also include the episodic regimes here, too. If I understand correctly, this should work by adding the standard deviations, right?

Reply: This is an important point, which we take very seriously. We have significantly extended and improved our classification scheme of tectonic regimes. Based on plateness-mobility scatterplots, we define the P-M ranges that are covered by the three non-episodic “base” regimes: mobile, stagnant and sluggish/squishy (Fig. 4; Table 1). We then classify the six main regimes based on the time fraction they spend in each of these base regimes (Table 2).

- Line 231++: I agree that it make sense to study "hot" and "cold" scenarios, but you should mention already here what this may means in terms of planetary age: hot=young, cold=old. Apart from this, the range of R_h values look fine, but is $T_{CMB}=4250$ K really a good value for a hot/young planet? It seems more like a high-end value for Earth's present CMB temperatures. Also, since you discuss planetary pathways later on, you should have considered a “future” case with R_h and T_{CMB} lower than today. You should already highlight here, that you co-vary them, so there are not independent. This opens up the question, which one is more important though (I am pretty sure it is R_h)!

Reply: We now clarify that we co-vary R_h and T_{CMB} and that they are related to planetary age already at the beginning of the “Parameter Study” section. Ideally, we would further expand the parameter space (even though our coolest cases are already relevant for future Earth). Nevertheless, it should be fine to discuss implications that go beyond the explicit findings of the study. Recent estimates for present-day CMB temperature are significantly lower than 4250 K (e.g., Nomura et al., 2014), also because such temperatures would lead to widespread deep-mantle melting (e.g., Andraut et al., 2014).

- Line 233: A rheology with a yield stress is always non-Newtonian, isn't it?

Reply: This is correct. We are referring to the viscous part of the visco-plastic rheology. Therefore, we added “of our viscous rheological treatment”.

- Line 235: This statement, which you refer to in multiple places in your study, is based on findings from the 1980s with very simple convection models (e.g. no yielding etc.). I do not think it has ever been confirmed for state-of-the-art models like the one used here. There has been some work looking at this (e.g., Arnould et al., 2023, GRL and also Lenardic and co-workers 2019?), but how the inclusion of dislocation creep matters is not really understood yet, I would say. Therefore, I suggest you rephrase your statement here accordingly.

Reply: The approach pioneered by Christensen is widely used, and not only based on simulations, but also on boundary-layer theory. The base of the lithosphere becomes weaker for lower activation energies (Ea), and therefore, the boundary layer is thinner, similar to the effects of dislocation creep. Van Hunen et al. (2005) show that models with Newtonian rheology and low activation energy can account for realistic thickening of the oceanic plates with dominant dislocation-creep rheology.

Clearly, the low- Ea approach is not the best approach to model realistic composite rheology with diffusion and dislocation creep. We are forced to use it based due to computational limitations. And it is fair to say that the effects of this approximation are not yet fully understood in visco-plastic models with plate-like behavior. This is one reason why we explore Ea in the Suppl. Inf. (the other main reason being that results are sensitive to this parameter). In fact, these model results contribute to our understanding of the effects of Ea in visco-plastic models.

To address this comment, we reworded the statement, such as removing the reference to the Christensen paper (and related discussion) at the beginning of the “Parameter Study” section. In turn, we leave this discussion further down in the main text, where the results of models with different Ea are briefly discussed. There, we allow more space for the discussion of the theoretical backdrop of the low- Ea approach, also mentioning that grain-size dependence of viscosity may justify the use on an adjusted Ea . We also remind the reviewer that in our main model series, we did not reduce experimental values for Ea by a factor of 2-3 as recommended by Christensen, but we applied a less extreme adjustment as a conservative choice.

- Line 296++: I was really excited when I saw the title of this section, but rather disappointed after reading it. Frankly, I am not convinced that the numbers you provide

here are quite transferrable to other studies (which you already question yourself). Also, how you defined the limited values is not sufficiently clear. Do not get me wrong, Figure 5 still adds value, but unless you want to elaborate on it, I suggest incorporating this short section and the figure into the previous sections.

Reply: We agree with the reviewer. First, we removed this section and incorporated it in the previous section. Second, we completely overhauled and strengthened the classification of regimes (see e.g. Table 1-2 and Fig. 4). We think that the new classification may be more transferrable, but do not make any specific claims in this regard in the manuscript.

- Figure 3: Please add to the caption that the lowers panels are just zoom-ins to highlight the small values obtained in the stagnant lid regime and the PSL regime ... I actually wonder if the differences come out better when plotted in log-scale? ... It may be me, but I didn't find it too easy to distinguish between pale orange and pale red in the bottom panels. Please consider a different color combination.

Reply: We removed this Figure. We use a sqrt-scale for Mobility on the new Fig. 4, which replaces this according to our new classification scheme. In this new Figure, the colors are separated between panels, so we hope that this resolves the issue around colors.

- Line 325: "in many previous studies" A good place to cite a few of those.

Reply: Added.

- Line 330: Indeed, this is not clear, but you should try to discuss this in more detail as it is one of the key questions. There is an interesting work by Stern et al. (2018), *Geoscience Frontiers* 9, 103-119, which may be useful for this (see their Fig. 3). In this context, the question about how to transition from one "stable" regime to another is relevant. Going from a mobile/episodic lid to a stagnant lid is typically thought to be easier than going from a stagnant lid back to a mobile lid, because to hysteresis. See e.g. works by Lenardic & Weller, or Weller & Kiefer 2020

Reply: In this paragraph, we make the point that it (i.e., how the strength of the lithosphere should evolve with temperature or over time) is not obvious from a rock-

physics viewpoint. Later on, we discuss this further, citing the Stern et al. (2018) paper. We emphasize that the geological record for Earth is more consistent with the evolution in Fig. 6a, which shares some similarities with that depicted in Fig. 3 of Stern et al. (2018). In terms of switching back-and-forth between mobile and stagnant (or squishy) behavior, our calculations which exhibit episodic and episodic-squishy behaviors clearly demonstrate that switches in either direction are feasible and rather frequent, possibly facilitated by magmatism (even though we do not make this claim explicitly in the paper). We now cite Lenardic & Weller newly in lines 163-164.

To summarize this reply, we now discuss this issue in more detail, but the discussion is not fully contained in the paragraph the reviewer is referring to.

- Line 334: I am not quite sure what this is supposed to say. You mean a quasi-steady state is reached after 1-2 Gyr IN YOUR MODELS? This is highly dependent on the initial conditions and other things.

Reply: Thank you for your thoughtful review and valuable suggestions regarding our manuscript. We have reworded this for clarification. Our models typically reach the same tectonic regime as in the statistical steady state within 1-2 Gyr. They do not strictly reach the statistical steady state after this time. We also tested the role of the initial conditions, by varying the initial geotherm.

Specifically, regardless of the initial conditions (for example, different initial potential temperatures (T_p) settings such as $T_p = 1700$ K, 1900 K, and 2100 K in Model reference6 (ESL)), all models achieve the same final tectonic regime within 1-2 Gyr. This tectonic regime then remains stable over the subsequent 6 Gyr. As a conservative choice, we probe the statistical steady state only after 6 Gyrs onward.

The new supplementary figure (also included below) clearly illustrates how, under different initial conditions, the models converge to the same final tectonic regime within the initial 1-2 Gyr and maintain stability throughout the subsequent simulation period.

- Paragraph starting in line 345++: This paragraph adds indeed interesting and relevant discussion. To improve it further, I think the authors should touch upon the interplay of interior and atmosphere evolution (without too much detail, of course): Tectonic activity modulates outgassing activity, which can alter atmospheric composition and eventually change surface temperature, which feeds back into crustal rheology, ... There is work by Gillmann et al. 2014, JGR about this for example, even though somewhat specific for Venus ... Another aspect I would like to see addressed in this discussion is that the assumption of one global value for the YS is probably not very realistic (even if changed through time). Deformation can be highly localized and can also occur more likely in specific regions, determined by initial conditions or other processes. This could lead to a spatially heterogeneous yield stress which can trigger very different evolutions than some averaged uniform value. Some examples of causes: specific insolation patterns (Mercury, see Tosi et al. 2015), giant impacts causing hemispheric scale crustal dichotomy (Mars, many works), and for the Earth processes like dehydration stiffening (e.g. Capitanio et al., 2019) and the formation of continents, which add an irreversible crustal structure, likely accompanied with a spatially

heterogeneous effective yield stress (e.g. Rolf & Tackley 2011).

Reply: We appreciate the reviewer's insightful comments, which have greatly improved the clarity and depth of our manuscript. In response, we have revised the text (see lines 390-397 and 413-415) to address the interplay between interior and atmospheric evolution, and the role of lateral lithospheric heterogeneity, albeit focusing on Earth in this part of the manuscript.

- Line 365: I think this could be written in a better way. Here, one can get the impression that it is solely the YS that controls the evolution, but it is actually the contrast between the YS and the stresses YL imposed on the crust and lithosphere, that matter. If $YS \ll YL$, a stagnant lid is expected, if $YS \geq YL$, a mobile regime occurs. In other words, when the imposed stresses increase more than the YS through planetary evolution, the result would still be some mobile regime even though YS may mildly increase. Or, if YL decreases faster than YS through time, the result would be a stagnant lid, even though YS may decrease. While effectively we may arrive at the same point, I think this small difference is still important and should be made clearer in this discussion.

Reply: Thank you for your valuable feedback. We now clarify that it is the balance between dynamic stresses and the yield stress that controls lithospheric deformation (lines 416-423).

- Line 371: Cite Stern et al. 2018 here (your ref #44)

Reply: Added.

- Line 396/397: Can we be sure that the intrusion-extrusion ratio was always the same? Perhaps in the early Earth, magmatism was much more extrusive because the crust was initially thinner? I think, it is a bit too early to exclude the heat pipe at least for the Hadean. It seems to be a place where the heat pipe mode may still be active even today ...

Reply: We agree with the reviewer and removed the sentence that argues against the heat-pipe.

- Line 456+: It can help, yes. But you should also mention somewhere that this map explores just a tiny bit of the parameter space, and that there may be more regimes hidden somewhere and multiple ways to transition from another.

Reply: We added this caveat near the end of the main text.

Methods

- Line 491/492: I am honestly surprised to about this sentence. Many studies have claimed that mantle viscosity is a very important parameter. After all, it determines the Rayleigh number and thus the thickness of boundary layers, i.e. the lithosphere. In a model, with depth-dependent YS , the lithospheric thickness should be important for the effective strength of the lithosphere. I have a hard time with this statement and I suggest that the authors double-check it, and consider deleting it.

Reply: Deleted. We understand the crucial role of mantle viscosity in controlling the thermodynamic and mechanical behavior of the lithosphere, particularly in its impact on the Rayleigh number and lithospheric thickness. We fully agree with your point that η_0 plays an important role in the effective strength of the lithosphere, especially when considering models with depth-dependent YS . In our study, the primary focus is on investigating the impact of activation energy (Ea) in the upper mantle on the viscosity profile. Compared to η_0 , we believe that Ea in the upper mantle controls the viscosity contrast between the lithosphere and asthenosphere, and thus has a more direct influence on the viscosity profile and tectonic style, which is why we have prioritized the variation of Ea in our study. That the influence of η_0 on tectonic style is second-order is consistent with Lourenço et al. (2020).

SI Material

- Line 39/40: The values for Rh seem wrong here. Missing factor $1e-12$?

Reply: Fixed.

- Line 100++: If you have not tested the resolution for this regime, you should just say to and do not give untested speculations.

Reply: We added a high-resolution case (Rh125cc120_h) that is in the parameter space of the Episodic Lid, and the result meets the classification criteria for EL (Tables 1-2). Overall, all high-resolution cases that we run confirm that the main conclusions (including the first-order topology of the regime diagram) remain robust.

References

- Andrault, D., Pesce, G., Bouhifd, M. A., Bolfan-Casanova, N., Hénot, J.-M., & Mezouar, M. (2014). Melting of subducted basalt at the core-mantle boundary. *Science*, *344*(6186), 892–895. <https://doi.org/10.1126/science.1250466>
- Andrault, D., Monteux, J., Bars, M. L., & Samuel, H. (2016). The deep Earth may not be cooling down. *Earth and Planetary Science Letters*, *443*, 195–203. <https://doi.org/10.1016/j.epsl.2016.03.020>
- Hunen, J. van, Zhong, S., Shapiro, N. M., & Ritzwoller, M. H. (2005). New evidence for dislocation creep from 3-D geodynamic modeling of the Pacific upper mantle structure. *Earth and Planetary Science Letters*, *238*(1–2), 146–155. <https://doi.org/10.1016/j.epsl.2005.07.006>
- Lourenço, D. L., Rozel, A., & Tackley, P. J. (2016). Melting-induced crustal production helps plate tectonics on Earth-like planets. *Earth and Planetary Science Letters*, *439*, 18–28. <https://doi.org/10.1016/j.epsl.2016.01.024>
- Lourenço, D. L., Rozel, A. B., Gerya, T., & Tackley, P. J. (2018). Efficient cooling of rocky planets by intrusive magmatism. *Nature Geoscience*, *11*(5), 322–327. <https://doi.org/10.1038/s41561-018-0094-8>
- Lourenço, D. L., Rozel, A. B., Ballmer, M. D., & Tackley, P. J. (2020). Plutonic-Squishy Lid: A New Global Tectonic Regime Generated by Intrusive Magmatism on Earth-Like Planets. *Geochemistry, Geophysics, Geosystems*, *21*(4). <https://doi.org/10.1029/2019gc008756>
- Nomura, R., Hirose, K., Uesugi, K., Ohishi, Y., Tsuchiyama, A., Miyake, A., & Ueno, Y. (2014). Low Core-Mantle Boundary Temperature Inferred from the Solidus of Pyrolite. *Science*, *343*(6170), 522–525. <https://doi.org/10.1126/science.1248186>
- Stern, R. J., Gerya, T., & Tackley, P. J. (2018). Stagnant lid tectonics: Perspectives from silicate planets, dwarf planets, large moons, and large asteroids. *Geoscience Frontiers*, *9*(1), 103–119. <https://doi.org/10.1016/j.gsf.2017.06.004>

Tackley, P. J. (2000). Self-consistent generation of tectonic plates in time-dependent, three-dimensional mantle convection simulations. *Geochemistry, Geophysics, Geosystems*, 1(8), n/a-n/a. <https://doi.org/10.1029/2000gc000036>

Dissecting the puzzle of tectonic lid regimes in terrestrial planets

Tianyang Lyu, et al.

(Response to the reviewers' comments)

REVIEWER COMMENTS

Reviewer #1 (Remarks to the Author):

Frankly, I'm not convinced by the authors' replies and the changes they made in the revised version of the MS. Here are the details.

1. I agree that analyzing the evolution of Earth-like planets using a steady-state approach is easier than doing that for transient evolution. However, that does not change the fact that the evolution of the planets is essentially a transient process, and, in my opinion, the authors do not show that their “steady-state” approach is indeed adequate.

Reply: The reviewer raises a crucial point. We agree that planetary evolution is fundamentally a transient process. Our choice of a steady-state approach was not a matter of convenience, but a matter of choice, in line with our study design and model philosophy/approach, primarily to allow rigorous analysis of model predictions (for transient models, such rigorous analysis is virtually impossible). But before elaborating on this (see below), we would like to try to convince the reviewer(s) that the steady-state approach is relevant for the transient evolution of terrestrial planets, which is at the core of the reviewer's concern.

Every model case (in the steady state) can be interpreted as a stage in planetary evolution (of course, this is a simplified perspective, but we prefer to make a simplification here in order to allow rigorous analysis of the model predictions). For example, cases with high internal heating and high T_{CMB} can be viewed as early (hot) stages. Cases with variable yield stresses can also be interpreted in that way, depending on how the effective strength of the lithosphere has evolved over time. The advantage of this approach is that different evolutions of e.g. the effective lithospheric yield stress (YS) over time, i.e. with respect to T_{CMB} /internal heating, can be considered. This is all discussed in detail in the last section of the manuscript.

Now, of course, the entire approach would fall apart IF regime transitions were significantly delayed in transient planets (and we suppose that this potential issue sparked the reviewer comment). Or even if hysteresis were important (i.e., a transient planet can assume different tectonic regimes for the same parameters depending on small changes in initial condition; so it's all a bit unpredictable). However, we now clearly show that neither transitions are

significantly delayed, particular along the tectonic paths through the regime diagram discussed in the manuscript, nor hysteresis is important.

We demonstrate this by running additional models, now presented in the Suppl. Inf. Text S4 and Figures S7-S8. These models are not strictly steady state. We initialize (or “restart”) models from the predicted temperature and compositional fields of steady-state models, but with different parameters as in said model. Hence, this test is accomplishing two things: (A) It tests what happens during a transient stage, i.e. during a stage in which the planetary thermal state or the surface yield stress abruptly change. (B) It also allows a comparison of two different models with the same parameters but different initial conditions (first, the initial condition at timestep zero, simplified radially-symmetric temperature field without compositional heterogeneity; second, the predictions of a steady-state model with different parameters than the restarted case after 10 Gyrs model time). In terms of (B), this tests whether significant hysteresis occurs in our model setup. In terms of (A), this addresses the issue of transient evolution of planets in an extreme end-member case, i.e., for abrupt (=immediate) changes of parameters. Real changes of core temperature, internal heating or lithospheric behavior are (likely) much more protracted, occurring over billions of years.

These experiments reveal two key findings:

1. As the parameters are abruptly changed, the plate-mantle system readily evolves towards the new regime that is expected for the new parameters according to our regime diagram(s). This is particularly the case when changing YS from higher to lower values (which is at the heart of our discussion of Earth’s tectonic history, see below). The new regime is assumed almost instantaneously. But we also when changing YS from lower to higher values, the new regime is readily assumed (within 0.1~1.5 Gyrs) as expected.
2. That the same tectonic regime is readily assumed at any given parameters, i.e. for the “original” cases presented in the main text AS WELL AS the restart experiments in Fig. S7-S8 (as expected from our regime map), also demonstrates that hysteresis is not a significant issue. Only “weak” tectonic regimes (as assumed at low YS) display a (somewhat limited) self-stabilizing effect relative to “strong” regimes, but only for up to 1.5 Gyrs (true hysteresis would/can be forever).

We now discuss the results and implications of these new tests in the MS (lines 460-471 in the main text), and in more detail in Suppl. Text S4.

In response to the other part of the comment: Modelling steady-state planetary evolution is not “easier” than modelling transient planets. Modelling transient planets is just as straightforward as modelling steady-state planets. The latter has been routinely done with the modelling code applied (“StagYY”) (Armann and Tackley, 2012; Jain et al., 2019; Lourenço et al., 2020; Nakagawa and Tackley, 2010, 2004; Tian et al., 2023). We would simply need to switch on core cooling and time-dependent internal heating, which are standard settings in StagYY. We made a deliberate choice to focus on steady-state models based on our scientific philosophy in designing the study. Our scientific question in the beginning was to identify possible tectonic transitions and thereby inform our understanding of planetary evolution.

In particular, the steady-state approach allows us to analyze the regime assumed by each case in a rigorous statistical manner. Our detailed quantitative analysis for identifying tectonic regimes is explicitly praised by Reviewer #3, and a novel result in itself, one that will hopefully set new standards for future studies of this kind. It also enables us to put the regimes in direct relationship to each other (i.e., to map them out in a regime diagram, which is the main objective of the study, as reflected in the title), and thus to build a robust, quantitative framework of tectonic behavior of the mantle-plate system on rocky planets.

2. The authors are excited to introduce a novel tectonic regime for early Earth evolution that combines elements of previously proposed squishy-lid and episodic regimes. Naturally, many transient tectonic regimes might exist between heat pipe, stagnant, squishy, and mobile-lid regimes. As a result, identifying one that appeared in earlier modeling studies (as authors also agree) but was unnamed is, to my mind, relatively insignificant.

Reply: In our view, this is an unfair comment. In the first letter, Reviewer #1 said that our results must be wrong because they are different from those of Lourenço et al. (2020). We replied that (1) there are differences in the setups of both studies and (2) that some cases in Lourenço et al. (2020) **might** (it is impossible to tell from their figures as they do not plot flatness for all their cases) be consistent with the episodic squishy lid (ESL), but that the ESL **might** have remained unidentified as a regime as such (we also emphasize that transitions from the mobile lid to the plutonic-squishy lid or vice-versa have not been explicitly discussed in their paper). Even if tectonic regime transitions similar to ESL **might** have occurred in their study (which entirely remains a speculation at this point), it would be extremely difficult to distinguish true ESL from the effects of transient evolution in their models (because they did not adopt a statistical steady-state approach, see response to first comment above). In their response to this, Reviewer #1 now essentially says that our study is insignificant because the results are not really new, which in our view is twisting our (perhaps too speculative) reply in an unfair manner.

Again, even if (which is not evident) a result similar to our newly-found ESL regime were buried somewhere in the results of Lourenço et al. (2020), this does not undermine the novelty of this aspect of our study. Identifying and exploring a result that has been previously unrecognized is not insignificant, on the grounds that it may have already been buried somewhere in previous studies. In analogy, Columbus' "discovery" of the Americas was not insignificant (for Europe) because the Vikings had already been there (obviously, the Native Americans had been there long before anyway; so, this analogy is meant from the perspective of Europe at the time of Columbus' voyage).

We also agree that it seems quite natural that ESL is between all three base regimes (mobile, sluggish/squishy, stagnant), the plutonic squishy lid (PSL) is between sluggish and stagnant lid regimes, etc.. However, these results are nevertheless novel. Putting all six identified regimes on a map in a comprehensive way has not yet been done. Showing that PSL is really an episodic regime between stagnant and sluggish/squishy is also new. And demonstrating the similarity

between the sluggish lid and the squishy episodes of PSL/ESL is novel, too. The relevance of the episodic-squishy lid (ESL) is underscored by its central position in our regime map, which spans a large swath of the parameter space. This is also consistent with geological evidence (e.g., Earth) and morphological evidence (e.g., Venus), as we now note in the main text (lines 525-528). If the final result at the end of the day feels natural and somewhat obvious, it demonstrates the elegance and significance of our study. Overall, we are convinced that our study contributes to further our understanding of the similarities and differences between the six identified tectonic regimes. For example, for a large part of the geodynamics community, even the difference between the sluggish lid and the PSL has been unclear so far. We are mapping out the regime transition between the two here for the first time.

Finally, we would like to emphasize the potential significance of ESL for understanding physical processes and the geologic record. As mentioned in lines 406-413, ESL may help to reconcile the debate about the onset age of mobile-lid tectonics (offering the intriguing possibility of multiple such onsets, interspersed by episodes of stagnant and/or squishy lid tectonics). As mentioned in lines 499-508, ESL may also help to reconcile the crater record on Venus with evidence for surface deformation.

Moreover, I would be careful in taking all these regimes very seriously based on the modeling of the type presented in this MS and other similar models. The reason is that in those models, the plastic strength of the lithosphere is assumed to be constant and homogeneous, a so-called von Mises type of plasticity. That is a strongly simplified approximation, which surely is wrong for the realistic lithosphere, where strength depends on normal stress, i.e., practically on depth (Mohr Coulomb friction). What is even more important is that in reality, the strength is very heterogeneous, being much lower at plate boundaries than inside the plates. While this simplified approximation played an important role in mimicking plate-like behavior, researchers still seem to forget about these models' limitations by seriously discussing the details of tectonic regimes based on such models.

Reply: The reviewer #1 is not correct in stating that our plastic yield strength applied is independent of pressure (see equation at lines 571-573). Our plastic yield strength (YS) has a pressure-dependence, albeit a relatively weak one.

In similar global studies, different approaches in terms of the pressure-dependence of the YS have been followed. Some studies have used a variable cohesion coefficient and fixed friction coefficient ($FC=0.6$), to approximate Byerlee's Law. The friction coefficient is practically the pressure-dependence of the YS. The disadvantage of this approach is that $FC=0.6$ may be unrealistically high. On a global scale, the strength of plate boundaries in the brittle regime may be essentially limited by dynamic friction, not static friction (Karato and Barbot, 2018), which imply a FC of ~ 0.1 . In other studies, such an approach has been amended by a so-called "ductile yield stress" with a usually higher surface value but very low pressure-dependence. This "ductile" yield stress approximates the poorly-understood behaviour of the lithosphere

near the brittle-ductile transition (see figure below, dashed line). Taking the brittle and “ductile” YS branches together, this yields a bi-linear approximation of the strength envelope.

[editorial note: third party material]

Fig. R1: Strength envelope of the oceanic lithosphere (Azuma et al., 2013; adapted from Kohlstedt et al., 1995).

To further simplify this bi-linear parameterization, we follow a third approach (analogous to Lourenço et al. (2020) and many other global-scale studies with plate-like behaviour). We disregard the brittle branch and only keep the “ductile” branch. The intersection of the “ductile” branch with the surface is the “effective surface yield stress”, i.e., one of the main parameters explored here. We note that it is the peak strength in the mid-lithosphere that controls the overall behavior of the plate (Kohlstedt et al., 1995). Indeed, it has been shown in many studies that changing the details of the strength envelope parameterization (e.g., varying the friction coefficient instead of the surface yield stress) does not change the overall global tectonic behavior (except for the specific parameter thresholds, at which regime transitions occur). Thus we prefer to keep our models simplified in this regard (Occams razor).

To address this comment, we briefly discuss the limitations of such an approach (lines 578-580), noting that this and similar approaches are the current state-of-the-art. Some studies, as pointed out by Reviewer #4, explore the implications of a more advanced damage rheology with grain-size evolution (and hence lithospheric memory), but this approach has not yet been applied to long-term global-scale mantle-convection models with mantle melting and self-consistent formation of crust and mantle heterogeneity. We stress (also discussed in the manuscript) the critical importance of melting and melt extraction (with combined intrusive melt emplacement and extrusion) for planetary tectonics (also see our reply to comment #1 of Reviewer #2), as also demonstrated by Lourenço et al., (2020, 2016). Future studies will have

to explore more complex (e.g., grain-size dependent) lithospheric/mantle rheologies in more detail, particularly in terms of their relevance to tectonic regimes with melting and melt extraction. Such an effort will require the exploration of large parameter spaces, since rheological and grain-growth parameters are poorly constrained (see e.g., Fuchs and Becker, 2022, 2021); hence, this effort is going well beyond this work.

3. In the ms. and in their reply to my comments, the authors paid much attention to their findings concerning scenarios for Earth-like planets moving from non-plate tectonic regimes to the plate tectonic regime. Therefore, I'll also focus on that issue here. Again, it is well established that decreasing lithospheric yield stress may result in a change from a non-mobile to a mobile (plate tectonics) tectonic regime. Therefore, a statement that the planet will move from a non-mobile regime to plate tectonics if its lithospheric yield stress decreases enough is trivial. It is more valuable to discuss possible mechanisms for how that could happen.

Reply: We totally agree with the reviewer. Evidently, the relationship between stagnant lid and mobile lid (and episodic lid in-between) with respect to yield stress is not new. We actually never highlighted that this would be the case, neither in the paper, nor in the reply, nor the cover letter.

That being said, many other aspects of possible regime transitions are new (see reply to first comment). The tectonic evolution of Earth discussed in the manuscript does not mainly rely on a transition from the stagnant to the episodic lid, and then to the mobile lid (plate tectonics). It incorporates transitions to additional regimes, including PSL and ESL, as well as the sluggish lid as a possible future regime (see Figure 6a and related discussion). We actually focus our discussion on possible transitions of Earth from the stagnant lid through the ESL regime, and then to the mobile-lid regime. Therefore, the main new result of our study (putting all 6 regimes, including ESL/PSL, on a regime map), is critical to have such a discussion. The relevance of a tectonic path that evolves through PSL/ESL is an important result for the understanding of geologic history. For example, debates in terms of when plate tectonics started, or in terms of which regime was dominant during the Archean/Hadean may be resolved by multiple episodes of mobile, squishy and/or stagnant lids within the ESL regime during Earth's history. We further highlight this implication by amending our discussion. Another new aspect of our study is that we find that the mobile lid occurs almost independently of thermal parameters (internal heating plus T_{CMB} , vertical axis of our regime diagram).

A relevant dynamic scenario was, for instance, suggested by Korenaga (2007, JGR), who proposed that the penetration of water through cracks due to the thermal cracking of oceanic lithosphere have lowered lithospheric strength and enabled plate tectonics already very early in Earth's evolution, just after the magma ocean stage. Sobolev and Brown (2019, ref 8 in the MS) suggested that the lithospheric strength was reduced due to the lubrication effect of continental sediments supplied to the oceans after the rise of continents, which enabled plate

tectonics after a kind of hybrid squishy-lid and plume-induced subduction tectonic regime in the Hadean and early Archean.

Reply: We thank the reviewer for these constructive suggestions, which could help to explain why the yield stress effectively decreased over time. We amend our manuscript to include these mechanisms (lines 444-446).

Those suggestions may be right or wrong, but the suggested mechanisms are specific. On the contrary, the authors suggest that the decrease in strength may be due to the Earth's cooling without pointing to the possible responsible mechanism. Their only argument is that many potentially relevant rheological mechanisms, such as Peierls creep or dislocation creep, are thermally activated. However, these mechanisms are slowing down with decreasing temperature and, therefore, cannot contribute to lowering the effective lithospheric yield strength.

Reply: The reviewer missed part of our discussion of rheological mechanisms. We thoroughly discuss two different mechanisms (since the first submitted version) which can account for an effectively decreasing YS over time (with Earth cooling): temperature dependence of dynamic friction (Karato and Barbot, 2018) and the competition between damage and healing (regulated by grain size) (Bercovici and Ricard, 2014). These two mechanisms have also been highlighted by Reviewer #4. It is possible that our discussion of these mechanisms was not sufficiently clear. To further clarify these issues, we largely rewrote this section and added more details (lines 420-444).

The authors suggest the paths that an Earth-like planet could follow through the different tectonic mechanisms during its evolution. Again, as in previous studies, the key changing parameter is lithospheric strength. The starting tectonic regime in the sequence of regimes in a decreasing strength path (Fig. 6a) is a stagnant-lid regime. Why that? In Korenaga's model, the regime is close to the mobile-lid from the beginning. In Sobolev and Brown's model, the starting regime is something close to the authors' episodic-squishy-lid regime and not the stagnant-lid regime.

Reply: Of course, this is a controversial issue, and we hope to contribute to this debate. Various models have been proposed regarding the initial tectonic regime (during the Hadean), most of which are conceptual and do not rely on quantitative mantle-convection simulations. Our arguments are grounded in our predictions of physically self-consistent models. According to our findings, there are just two possible types of paths through Figure 5b that end up in the PT regime in the present day (as long as we neglect very complex paths).

(A) paths vertically downward (constant YS over time) or downward right (increasing effective surface YS over time);

(B) paths downward left (decreasing effective YS over time).

Paths as in (A) always stay in the mobile-lid regime, inconsistent with most interpretations of the geologic evidence. Therefore, we favor paths as in (B). This has implications for the dominant mechanisms of lithospheric deformation during Earth's evolution.

Looking at this issue from a perhaps more neutral perspective, the largest uncertainty in terms of the initial tectonic regime of solid-state Earth mantle evolution (in the Hadean) is what happened at the very end of the magma ocean stage. In all our models (and all other evolution models [with core cooling, etc.] that we are aware of), there is at least a short initial phase with a stagnant lid. To some extent, this result might be promoted by the initial condition of the models, which is somewhat arbitrary. However, our study demonstrates that there may be some significance to this result. In a very hot and vigorous mantle, the stresses at the base of the lithosphere are small, and it will be difficult to disrupt/break the lithosphere. This physical mechanism (relevance of stresses from below) has been highlighted in previous works (e.g., Heck and Tackley, 2011; Stein et al., 2013), and is also emphasized by our additional parameter suite with variable activation energy (Suppl. Info.). We amend the discussion in the section about Earth evolution to highlight the uncertainty of initial condition and Hadean tectonics just after the magma ocean stage.

We cannot comment on "Korenaga's model" as the reviewer does not specify which of his papers is meant. In any case, we now cite Korenaga, JGR 2007 (as is mentioned by Reviewer #1 above). We also emphasize in the main text that the onset age of plate tectonics is highly debated (citing another Korenaga paper), and that the ESL regime may reconcile multiple "onset times" of PT.

The work by Sobolev and Brown, (2019) does not explicitly model early-Earth dynamics. It relies on regional subduction models with more or less sediments in the system and therefore focuses on the transition from Proterozoic tectonics to modern-style plate tectonics (both could be sub-regimes of the mobile lid). They also discuss the geologic evidence of Archean tectonics, which, as the reviewer points out, aligns with our interpretation of Earth history in section **Earth tectonic history with evolving lithospheric yield stress**. We added a citation to their work there.

That shows how speculative and controversial the regime's scheme, as suggested by the authors, is.

Reply: It is controversial, but not speculative. Our conclusions rely on quantitative model results that are based on basic physical principles. We further demonstrated their robustness by resolution tests and our new "restart experiments (Suppl. Text S4). We hope to address the existing controversy with our results, even though we are aware that the debate will continue to some extent. In the end, our results directly address at least three of the ten biggest questions in 21st century Geosciences, and indirectly a few more (<https://www.nationalacademies.org/news/2008/03/ten-questions-shaping-21st-century-earth->

science-identified). As mentioned above, the ESL regime may reconcile some hotly debated contrasting end-member models for early-Earth/Proterozoic global tectonics with implications for the dynamics and evolution for the entire Earth system.

Finally, the resolution of the presented models is not impressive either. The 256 horizontal cells in (I guess) half annulus model means cell width about 78 km compared to about 20 km cell width in the recent model with the similar modeling technique (Jain et al, 2022, Front. Earth Sci.).

Reply: The reviewer is correct that our models are half-annulus.

We refer the reviewer to our resolution test in the Supplementary Information. In the high-resolution test, we used a 2D quarter-annulus domain with a resolution of 192 (horizontal) \times 96 (vertical) cells. This corresponds to an average horizontal resolution of \sim 50 km and a vertical resolution of \sim 30 km, with the resolution near the surface reaching \sim 18 km.

While the model by Jain et al., (2022) used a finer grid, their study explored only 38 cases in total. In contrast, our study systematically investigates a substantially larger number of cases, which necessarily places stricter constraints on achievable resolution due to computational limitations. Nonetheless, our tests clearly show that this resolution is sufficient, and that the main physical features are robustly captured (Text S2).

We also note that other published global-scale modelling studies on tectonic regimes such as the plutonic-squishy lid (Lourenço et al., 2020, 2018; Rozel et al., 2017) used a similar resolution (512 \times 64 grid cells in a full-annulus (i.e., global) configuration). We emphasize that the vertical resolution near the surface, and not the horizontal resolution, is particularly important for accurately capturing lithospheric deformation, and our models are well-resolved in this regard (partly due to grid refinement near the surface). The grid refinement near the top and bottom boundaries of the domain is now clearly explained in Text S2.

Reviewer #2 (Remarks to the Author):

The majority of my comments and questions have been answered with the revised version of this manuscript. However, a few of my comments on the original submission I don't think were fully answered in the review response or revised manuscript, probably due largely to some miscommunication on exactly what my points of criticism were. I will describe those here below. None of my comments requires extensive new modeling or reworking of the manuscript (I don't think any new modeling is needed at this stage). They really just concern some of the specific wording and claims being made. As a result, I think only minor additional revision is needed before the paper can be accepted.

Remaining comments:

1. The revised manuscript is still claiming that running models to statistical steady-state is solving the problem of hysteresis (e.g. lines 95-102). But just running models to statistical steady-state does not get around this issue. In Weller's work that I cited in the previous review, they documented cases where two models could have the exact same input parameters but end up in different tectonic regimes *at statistical steady-state based on initial conditions*. The different initial conditions are comparing starting from a fully developed mobile lid state to starting from a fully developed stagnant lid state, and Weller & Lenardic found that they needed much lower yield stresses in some cases to get mobile lid convection to result with a stagnant lid initial condition than with a mobile lid initial condition. Changing the initial mantle temperature, as was done in the supplement, is not a full test of the effect of initial conditions in the way Weller & Lenardic did.

I don't think this manuscript needs to do such a full test; I just don't think it should claim running to statistical steady-state avoids issues with hysteresis. The results of this manuscript are still interesting even with hysteresis remaining as a caveat.

Reply: We sincerely thank the reviewer for the insightful comments regarding the issue of hysteresis. As pointed out by the reviewer, running models to a statistical steady-state alone does not fully eliminate hysteresis. We are well aware of previous studies, such as Weller & Lenardic (2012, 2018), which have demonstrated that models with identical parameters but different initial conditions—e.g., starting from fully developed mobile lid versus stagnant lid states—may converge to different tectonic regimes, and that hysteresis can significantly affect the results.

In our work, we have made efforts to explore the impact of hysteresis in several ways, and we would like to clarify our stance as follows.

We have run additional “restart” experiments, in which key parameters (surface yield stress and upper-mantle activation energy) are modified upon restart. In other words, a case is “re-started” from the final solution of a case with different parameters (i.e., it is restarted from a

case with a different tectonic style than expected to be assumed with the new parameters). We have run ten such restart experiments.

So, any given restart experiment and the corresponding “original” case with the same parameters have been run from different initial conditions (in the former case, the predictions of another model; in the latter case, radially-symmetric initial conditions with random noise), but governed by the same parameters. We have run the restart experiments to address reviewer comments about the transient evolution of planets (R1 and R4), but these cases also address the issue of hysteresis. Suppl. Text S4 and Figures S7-S8 describe the results of these additional cases. The paragraph in lines 462-471 in the main text summarizes these results (also see our reply to comment #1 of Reviewer #1).

Thus, there is no evidence that hysteresis plays a dominant role in our models. These contrasting conclusions (w.r.t. Weller & Lenardic) are in our view well explained by the absence of mantle melting and crustal production in their models. However, as we do not explore this issue in extensive detail in our study, we do not question their conclusions explicitly. We just mention (based on our additional tests) that hysteresis is not a big issue in our models.

2. Lines 65-66 say that the sluggish lid regime is sometimes called the ridge only regime. I don't quite agree because as I said before this implies that sluggish lid and ridge only are interchangeable terms. When in my opinion they are describing different things; I would place ridge only as a subset of sluggish lid, but not all sluggish lid is ridge only.

Reply: We agree with the reviewer. In response, we have removed the term "ridge-only regime" to avoid any potential confusion and to ensure more precise terminology in the revised manuscript.

3. Line 164: I've never heard anyone call episodic lid behavior “hysteresis.” Instead, I've heard hysteresis used to describe what I said in point 1; models that go to different end states due to different initial conditions despite the same parameters.

Reply: We have removed this sentence.

4. Lines 174-175: I still don't get why the sluggish lid has a very high plateness. The revised text is clearer than the original version, but still doesn't explain it. If the convergence zone is broad, that at least sounds like deformation is distributed across this wide zone, which should make plateness lower. I wonder if there is a figure that could be added (even just in the supplement is fine) helping illustrate how the broad convergence zone still leads to high plateness, when in my mind high plateness requires narrow, localized deformation zones.

Reply: We thank the reviewer for the thoughtful comment regarding the plateness in the sluggish lid regime. As shown in the newly added viscosity snapshot in Figure S9, although the convergence zones are broad, the deformation is highly localized in narrow, V-shaped plate-scale yielding zones characterized by low viscosity and high strain rates. These focused zones of deformation dominate the surface strain pattern and effectively maintain sharp plate boundaries. Thus, the broad convergence zones are composed of two localized high-strain regions (and at face value, a stable micro-plate in-between – even though double-sided subduction may be an artifact of our models (see lines 172-176)), which preserve a high overall plateness. Figure S9 has been included in the supplementary materials and is referred to in the main text (line 174) to better illustrate why wide convergence zones can still correspond to high plateness.

5. Line 341: Are small and large grain sizes here flipped? Hot upwellings should lead to larger grain sizes due to high temps and rapid grain growth, with cold downwellings having smaller grains due to sluggish grain growth.

Reply: We thank the reviewer for catching this typo. Yes, these have been flipped. Now corrected.

6. Lines 356-368: I have no objections to this paragraph, but it isn't really answering the comment in my first review that sparked this. In my original review I was asking about Fig. S1, where we see that below E_{UM} of about 100 kJ/mol, there is no stagnant lid. It makes sense that at some point E_{UM} is low enough that even with no yielding or other weakening, it still will not produce a stagnant lid. Viscosity has to be about 10,000 times larger at the surface than the mantle interior to produce stagnant lid convection, as shown in Solomatov, 1995; otherwise there will be something like sluggish lid convection. All I was asking here originally is if we were seeing that transition at low E_{UM} in Fig. S1 - do we not see stagnant lid convection at low E_{UM} in Fig. S1 just because the viscosity variation is too low to ever produce a stagnant lid?

Reply: We thank the reviewer for this insightful comment. We analyzed the low E_{UM} cases and confirm that the viscosity contrast between the cold surface and the hot asthenosphere is indeed greater than a factor of $\sim 10,000$. This finding confirms that an insufficient viscosity contrast is not the reason for the persistent mobile-lid behavior in these models. To make this explicit in the manuscript, we have added a sentence at the end of the relevant paragraph (at SI lines 63-65).

7. Line 399: To be pedantic, grain size reduction in the Bercovici damage theory is not driven by stress alone, but by the deformational work, product of stress and strain-rate.

Reply: We thank the reviewer for highlighting that, in the Bercovici damage framework, grain-size reduction is driven by deformational work (i.e., the product of stress and strain-rate), rather than by stress alone. We have updated the main text (line 425) to reflect this more precisely.

Brad Foley

Reviewer #3 (Remarks to the Author):

The authors have done an excellent job in revising their original draft. I see that all of my comments and concerns have been very satisfactorily addressed. I am particularly please about the fact that the authors have put great effort into improving and/or adding several figures to the revised version, and that they have tried to formalise the method to assign a tectonic regime to each case, based on transparent and quantitative criteria. While details of this method can certainly be discussed and improved in future studies, this approach is a good example for the community about how tectonic regimes should be analysed.

Reply: We thank the reviewer for their thoughtful feedback.

I only have three, very minor comments on the revised version. These can be easily added in the proof-reading stage after acceptance. Otherwise, I will be happy to see this work published soon.

(1) Line 79: „realistic ratios of intrusion to extrusion“: Given that this ratio is quite uncertain and rather debated, it would be good to add one or two references here.

Reply: We thank the reviewer for their valuable comment. We have added two relevant references as suggested to support this statement.

(2) Line 102: „predict the tectonic history of rocky planets“: given the assumptions (like the statistical steady-state), I do not think your models are PREDICTIVE in terms of tectonic history yet, but rather SUGGESTIVE. Consider rephrasing here.

Reply: We thank the reviewer for their suggestion. We have reworded this section removing the verb “predict” w.r.t. to the tectonic history of terrestrial planets. Overall, the paragraph is now a bit more specific, and more representative of our actual results. We now finish the paragraph with this half-sentence: “[...], with implications for the tectonic history of rocky planets.”

(3) Line 174: You seem to have added a phrase about double-sided subduction here. I do not think this is necessary at all, but if you add it, you should also add a half-sentence that „this may not be representative of present-day subduction on Earth and is probably at least partially caused by our relatively simple rheological model“.

Reply: We have added a caveat sentence very similar to that suggested by the reviewer, and a citation to support this.

[editorial note: reviewer name redacted]

Reviewer #4 (Remarks to the Author):

I've read the paper and the first set of reviews, and also the 2nd review from Referee 1.

Admittedly I'm not a giant fan of such parameter sweeping studies that are sort of a random fishing expedition posing as hypothesis testing. But there seems to be at least more positive response from the other referees. While I mostly agree with Ref 1 that his/her comments were not adequately addressed, I think the authors could address the concerns with a little more thought.

1. Using statistically steady state solutions to elucidate evolution and transitions between states has limitations. It can tell us something about the state on either side of a transition but very little about the transition itself (how long it takes for example). Arguing that it's easier to do steady state is not a good rebuttal. But arguing that bracketing transitions is useful (i.e., articulating where you are to where you end up is just as important on a roadmap as an estimate of how long it will take you to get there). So I think there is still utility, but the authors should make a better case.

Reply: Running statistical steady-state models with StagYY is not easier than running evolution models. Analyzing them more straight-forward, and that's exactly why we chose to run steady-state models. In fact, rigorous analysis of evolution models in the context of distinguishing episodic from non-episodic regimes is virtually impossible (any regime transition may be either due to transient evolution of basal/internal heating, or a true episode – it is impossible to tell). The steady-state approach allows us to systematically distinguish regimes and establish quantitative statistical methods to do so (Figs. 4-5; Tables 1-2). This is a pre-requisite to robustly map out regimes. Hence our choice to focus on steady-state models (this rationale is now also more clearly described in the last paragraph of the introduction section). See also reply to Reviewer #1 comments.

Our new statistical methods to quantitatively distinguish regimes (also praised by Reviewer #3) will hopefully set a new standard for global-scale mantle-convection models with plate-like behavior, and will thereby help to better understand the dynamics of the plate-mantle system, including in upcoming studies with more complex lithospheric rheology.

2 and 3 Combined. First, yes I agree that finding a new regime that isn't really new, just not named yet, is not terribly novel.

Reply: We disagree that the episodic squishy lid (ESL) is not really novel. It is an episodic regime that has not yet been found or described. It is different from the plutonic-squishy lid described in Lourenço et al. (2020) in that it exhibits rather long mobile episodes in addition to squishy and near-stagnant episodes. In a way, any episodic regime is a transitional regime with respect to previously discovered regimes, but episodic regimes are still highly relevant (e.g., the classical “episodic lid” has been hotly debated for a long time in terms of its applications to Venus; the ESL regime might be relevant for multiple onsets of mobile-lid behavior over

planetary history). The central location of the ESL regime in our regime map, covering a large portion of the parameter space, further highlights its importance. Additionally, its relevance is supported by both geological (Earth) evidence and morphological (Venus) evidence, as added in the last paragraph of the main text. Also see our more detailed reply to comment 2 of Reviewer #1.

But putting into the context of other regimes and transitions, is not useless.

Reply: We agree with the reviewer. Putting all regimes in context to one another is perhaps the most novel aspect of the study, hence the title of the manuscript. For example, our study elucidates the similarity of the sluggish lid and of the squishy episodes in PSL/ESL. It shows that PSL is an episodic regime between the sluggish and stagnant etc. All these are new findings. It also maps out which regime transitions may occur in which parameter ranges, or which regime transitions are unlikely to occur because they do not share a boundary in the regime diagram.

But more to the point about the treatment of lithospheric yield stress: To be up front, I believe the pervasively used yield stress approach is fundamentally flawed. The yield stresses are always some unrealistically small value.

But more important (and as I and others like Mike Gurnis and most recently Fuchs and Becker have written about for 30 years) it fails a fundamental test. That is, a yield stress allows weakness to occur suggestive of a plate boundary, but as soon as the stress and deformation cease, the weak zone vanishes, which is not physical or even geological. Dormancy and inheritance of weak zones is well known and important for plate boundary activation and reactivation; and there are plenty of extant physical models to explain this (yes, grain-damage, but also others involving state variables like temperature, water content, anisotropy). The yield stress formulation is widely used because it's easy and it is what is made available in large codes (STAG and CITCOM), which many folks use and just more or less ignore all the arguments about why it is a faulty approach. Having gotten that off my chest, the current study is what is and perhaps, since it employs a steady-state approach, it can be argued that it side-steps the issue of weak-zone dormancy/inheritance which are strong temporal effects. And, just to try to be constructive and helpful, perhaps these effects might be used to answer some of the Referee's criticisms. First, I do agree with the Referee that arguing the transition from stagnant to mobile with a decrease in yield stress is trivial; it is indeed obvious but that's not necessarily fatal. But the reason why it decreases is indeed not explained in the paper and the Referee is justified in wanting the authors to providing some physical explanation about why the yield stress decreases with time. If the authors can't or aren't willing to address that, then I would consider that fatal. But I will throw a possible lifeline, or a few of them. There are a few physical mechanisms that might help them resolve this, and in fact the suggestions have been around for pushing 20 years already. It is true (as the Referee says) that the effect of cooling would at face value have a stiffening effect; and that has been known for a long time. It's the

same problem of why a planet with a cool surface like Earth has mobile lid (i.e., plates), while one with a hot surface like Venus does not. But that was the entire point of grain damage theories, going back to the papers of Landuyt and myself. and elaborated on with multiple papers by myself and others since then. That is, healing becomes less efficient with cooling, while the damage processes become more efficient; so the two both conspire to make damage (and lasting, dormant damage) prevalent on a cooling planet, but less so on a hot planet. That is one way the so-called or effective yield stress might decrease with time. But part of the same effect, or any effect that builds and accumulates dormant weak zones (not just mylonitization but fractures, hydration, anisotropic fabric), is that old unhealed scars progressively make the lithosphere weaker (Gurnis and Zhong would have made this case in the mid-1990s). Finally, Karato and Barbot (2018) offered a mechanism to imply that dynamic weakening is temperature dependent as a mechanism to explain the Earth-Venus dichotomy also. It's not that one of these is right and the others are wrong; they are all probably activated in some way. But the point is, if the authors took some of these microstructural and rock mechanics effects into account and didn't just rely on a faulty yield-stress formalism because so many others have (plurality is not the same as validity), they could probably strengthen their case.

Reply: We thank the reviewer for these suggestions to strengthen our paper. We agree with the reviewer that our long-term evolution models use a simplified lithospheric rheology (now explicitly clarified in the Methods, lines 578-580). This is partly driven by computational limitations, as we set out to run a rather large parameter study on a global-scale into the steady state (i.e., over 10 Gyr model time for each case), but mostly a deliberate choice to keep our models simple in this regard. Modelling is always modelling, and there is always a choice that needs to be made in terms of where to simplify the models. In turn, in several other aspects (geometry, melting, crustal formation, phase transitions, self-consistent physical properties), our models are quite complex/realistic.

We also agree with the reviewer that the planetary-scale effective lithospheric yield strengths (which are required for mobile-lid behavior) are always “some low value”, at least compared to lab rock deformation experiments of mostly-intact rocks. How to bridge this gap (how to upscale from the lab scale, both in time and space, to the lithospheric or even planetary scale) remains a big discussion in our field. Accordingly, we emphasize in several places in our manuscript that our applied yield stresses are “effective” (the word count of “effective” in the context of yield stress is ~20 in the main text of the manuscript). Possible mechanisms that can reduce the effective strength of plate boundaries or ductile shear zones are brought up by the reviewer (dynamic friction, “damage”).

Related to this, the reviewer makes two excellent suggestions in terms of how to justify decreasing “globally-effective” yield stresses over time (and with planetary cooling). These suggestions have been partly brought up because Reviewer #1 claims that we do not discuss any such physical mechanisms that could explain the decreasing YS over time. Just as an aside: Reviewer #1 is evidently wrong here. We have an elaborate discussion about exactly these two mechanisms in lines 370-427 of the last version of the manuscript. Essentially, the two mechanisms suggested by the adjunct Reviewer #4 have already been discussed in breadth in

our manuscript. We have discussed dynamic weakening (Karato and Barbot, 2018) and damage/healing (e.g., Bercovici and Ricard, 2014) starting from the first submitted version of the manuscript. We feel encouraged that an expert in exactly this field (Reviewer #4) suggests discussing these same mechanisms to strengthen our paper. Of course, it is entirely possible that Reviewer #1 missed these important points in our paper, because we have not explained them well. To further clarify and emphasize this issue, we largely rewrote and hopefully strengthened this section (now in lines 420-444).

Indeed, these two mechanisms, as the adjunct reviewer points out, are viable justifications to explain the decrease of effective YS over time (with planetary cooling). Our models are simplified in the regard that these mechanisms are not explicitly included. That being said, the modelled cohesion coefficient of any given case can be viewed as an “effective” property. Accordingly, a case towards the left side of the regime diagram can be interpreted as a planet with a lithosphere that has accrued more damage (in several places), or alternatively, as a planet with a cooler lithosphere, in which dynamic frictional instabilities are more efficient – compared to a case on the right side of the diagram. Future studies will be needed to study the effects of these rheological mechanisms more explicitly, perhaps also including the effects of continents on tectonic style. Our study certainly motivates such studies (last paragraph). The main advantage of modelling complex rheologies explicitly (as opposed to the “effective” approach used here) is that the locations of new plate boundaries, e.g. due to pre-existing damage, will be more realistically modelled. The main advantage of using an “effective” YS is that results are more generic and can be discussed, e.g., in the context of different lithospheric deformation mechanisms, or a combination of them (as we now do in lines 444-446). As the reviewer points out, such an approach aligns with our philosophy of using steady-state models.

On the numerical resolution comment. If it's just about being impressive, I don't see why that matters. The question isn't about how big your grid is, but whether it does the job and that the solutions are well resolved.

Reply: We agree with the reviewer. A resolution test is included in the Supplementary Information of our manuscript, demonstrating that the grid does the job.

[editorial note: reviewer name redacted]

Reference

- Armann, M., Tackley, P.J., 2012. Simulating the thermochemical magmatic and tectonic evolution of Venus's mantle and lithosphere: Two-dimensional models. *J. Geophys. Res.: Planets* 117. <https://doi.org/10.1029/2012je004231>
- Azuma, S., Katayama, I., Nakakuki, T., 2013. Rheological decoupling at the Moho and implication to Venusian tectonics. *Sci. Rep.* 4, 4403. <https://doi.org/10.1038/srep04403>
- Bercovici, D., Ricard, Y., 2014. Plate tectonics, damage and inheritance. *Nature* 508, 513–516. <https://doi.org/10.1038/nature13072>
- Fuchs, L., Becker, T.W., 2022. On the Role of Rheological Memory for Convection-Driven Plate Reorganizations. *Geophys. Res. Lett.* 49. <https://doi.org/10.1029/2022gl099574>
- Fuchs, L., Becker, T.W., 2021. Deformation Memory in the Lithosphere: A Comparison of Damage-dependent Weakening and Grain-Size Sensitive Rheologies. *J. Geophys. Res.: Solid Earth* 126. <https://doi.org/10.1029/2020jb020335>
- Heck, H.J. van, Tackley, P.J., 2011. Plate tectonics on super-Earths: Equally or more likely than on Earth. *Earth Planet. Sci. Lett.* 310, 252–261. <https://doi.org/10.1016/j.epsl.2011.07.029>
- Jain, C., Rozel, A.B., Hunen, J. van, Chin, E.J., Córdoba, A.M.-C., 2022. Building archaic cratonic roots. *Front. Earth Sci.* 10, 966397. <https://doi.org/10.3389/feart.2022.966397>
- Jain, C., Rozel, A.B., Tackley, P.J., 2019. Quantifying the Correlation Between Mobile Continents and Elevated Temperatures in the Subcontinental Mantle. *Geochem., Geophys., Geosystems* 20, 1358–1386. <https://doi.org/10.1029/2018gc007586>
- Karato, S.-I., Barbot, S., 2018. Dynamics of fault motion and the origin of contrasting tectonic style between Earth and Venus. *Sci. Rep.* 8, 11884. <https://doi.org/10.1038/s41598-018-30174-6>
- Kohlstedt, D.L., Evans, B., Mackwell, S.J., 1995. Strength of the lithosphere: Constraints imposed by laboratory experiments. *J. Geophys. Res.: Solid Earth* 100, 17587–17602. <https://doi.org/10.1029/95jb01460>
- Lourenço, D.L., Rozel, A., Tackley, P.J., 2016. Melting-induced crustal production helps plate tectonics on Earth-like planets. *Earth Planet. Sci. Lett.* 439, 18–28. <https://doi.org/10.1016/j.epsl.2016.01.024>
- Lourenço, D.L., Rozel, A.B., Ballmer, M.D., Tackley, P.J., 2020. Plutonic-Squishy Lid: A New Global Tectonic Regime Generated by Intrusive Magmatism on Earth-Like Planets. *Geochem Geophys Geosystems* 21. <https://doi.org/10.1029/2019gc008756>
- Lourenço, D.L., Rozel, A.B., Gerya, T., Tackley, P.J., 2018. Efficient cooling of rocky planets by intrusive magmatism. *Nat Geosci* 11, 322–327. <https://doi.org/10.1038/s41561-018-0094-8>

- Nakagawa, T., Tackley, P.J., 2010. Influence of initial CMB temperature and other parameters on the thermal evolution of Earth's core resulting from thermochemical spherical mantle convection. *Geochem., Geophys., Geosystems* 11. <https://doi.org/10.1029/2010gc003031>
- Nakagawa, T., Tackley, P.J., 2004. Effects of thermo-chemical mantle convection on the thermal evolution of the Earth's core. *Earth Planet. Sci. Lett.* 220, 107–119. [https://doi.org/10.1016/s0012-821x\(04\)00055-x](https://doi.org/10.1016/s0012-821x(04)00055-x)
- Rozel, A.B., Golabek, G.J., Jain, C., Tackley, P.J., Gerya, T., 2017. Continental crust formation on early Earth controlled by intrusive magmatism. *Nature* 545, 332–335. <https://doi.org/10.1038/nature22042>
- Sobolev, S.V., Brown, M., 2019. Surface erosion events controlled the evolution of plate tectonics on Earth. *Nature* 570, 52–57. <https://doi.org/10.1038/s41586-019-1258-4>
- Stein, C., Lowman, J.P., Hansen, U., 2013. The influence of mantle internal heating on lithospheric mobility: Implications for super-Earths. *Earth Planet. Sci. Lett.* 361, 448–459. <https://doi.org/10.1016/j.epsl.2012.11.011>
- Tian, J., Tackley, P.J., Lourenço, D.L., 2023. The tectonics and volcanism of Venus: New modes facilitated by realistic crustal rheology and intrusive magmatism. *Icarus* 399, 115539. <https://doi.org/10.1016/j.icarus.2023.115539>
- Weller, M.B., Lenardic, A., 2018. On the evolution of terrestrial planets: Bi-stability, stochastic effects, and the non-uniqueness of tectonic states. *Geosci. Front.* 9, 91–102. <https://doi.org/10.1016/j.gsf.2017.03.001>
- Weller, M.B., Lenardic, A., 2012. Hysteresis in mantle convection: Plate tectonics systems. *Geophys. Res. Lett.* 39. <https://doi.org/10.1029/2012gl051232>

Dissecting the puzzle of tectonic lid regimes in terrestrial planets

Tianyang Lyu, et al.

(Response to the reviewers' comments)

REVIEWER COMMENTS

Reviewer #2 (Remarks to the Author):

1. The manuscript now heavily appeals to physical processes like grain size evolution or thermal effects on earthquake rupture that are not included in their models to justify this path. The argument isn't very convincing since we don't know if the regimes would show up in the same way and in the same parameter space if the models actually did include these other effects.

Reply: We thank the reviewer for this feedback and appreciate the opportunity to clarify our reasoning. We acknowledge that our response letter may have become somewhat convoluted after multiple review rounds, but the main chain of arguments remains the same as in the first submitted version. Nothing has changed significantly in terms of this main red thread, and reviewer #2 has had no objections over several rounds of reviews regarding this.

Specifically, we disagree that our “manuscript now heavily appeals to physical processes like [...] to justify this path”. Our interpretation does not rely on these or any other physical processes (not included in the models). Our interpretation is based on the predicted regime diagram (our results) and on the geologic record (see next comment). The geologic record indicates that Earth evolved from a rather poorly constrained early tectonic regime through a plutonic-squishy or episodic-squishy or similar regime in the Archean to a mobile lid regime in the present-day. Based on our results, such a tectonic evolution requires effectively decreasing lithospheric strengths during planetary cooling over time. This is the chain of arguments to support one of our conclusions (summarized in Figure 6a), and we are convinced that this is robust and comes out of our results.

Now, based on this, we discuss potential physical mechanisms that could explain such effectively decreasing lithospheric strengths with planetary cooling // over time. We can't see that there is anything wrong in discussing the potential physical mechanisms for this intriguing result. In fact, reviewers #1 and #4 explicitly asked us to take this effort, and at least reviewer #4 is explicitly happy with the outcome. As reviewer #2 points out, future work should model mantle and lithosphere dynamics with more complex rheologies to test this explicitly.

Finally, and perhaps critically, we fully agree with the reviewer that we cannot conclude that the suggested mechanisms have indeed been critical or dominant for effectively decreasing

lithospheric strengths over Earth evolution. We do not want to imply that this would be the case. We now make this point clearly in lines 446-449. We have also reworded our manuscript slightly in several other places to clarify this issue further.

2. The manuscript also tries to appeal to observations to justify their preferred evolutionary path for the Earth. But despite stating that the manuscript will “discuss the geological evidence” “for decreasing lithospheric strengths over time” no actual observations of the early Earth are given. Instead, the manuscript discusses models (both conceptual cartoon models and physical models) that other studies have put forth. While the models in those cited papers are themselves based on observations (for the most part), they are particular interpretations of the observations. Appealing to other peoples’ interpretations rather than directly to observations themselves to support the argued for evolutionary scenario is unconvincing. Along the same lines there are repeated claims about how the regimes and evolutionary path the paper argues for can reconcile the geologic record, yet no specific observations are outlined nor is there any demonstration that the models can explain any key observations about the early Earth.

Reply: We fully agree with the reviewer that we can (and should) do a better job in terms of basing this discussion more on the original geologic literature (i.e., on geological observations). Such an effort should indeed strengthen our manuscript. We have revised the section accordingly, incorporating specific observations such as dome-and-keel structures in Archean cratons, widespread tonalite-trondhjemite-granodiorite (TTG) rocks, and Proterozoic indicators of horizontal motion (e.g., ophiolites and blueschists). These are now cited from foundational observational and geochemical studies (e.g., Sandiford et al., 2004; Moyer & Martin, 2012; Zhao et al., 2025). We have also adopted more cautious language (e.g., "consistent with" instead of "reconcile") to emphasize that our models align with these observations without overclaiming explanatory power. This enhances the manuscript's rigor and directly ties our tectonic-regime predictions to key geologic evidence.